# The Impact of IFN-γ Licensing on Mesenchymal Stromal Cells’ Mediated Immunoregulation and HLA Class II Expression: Emerging Evidence from In Vitro Results

**DOI:** 10.3390/ijms26199436

**Published:** 2025-09-26

**Authors:** Panagiotis Mallis, Theofanis Chatzistamatiou, Evangelia Gkatzoflia, Hava Zdrava, Eirini-Faidra Sarri, Efstathios Michalopoulos, Alexandros Spyridonidis, Catherine Stavropoulos-Giokas

**Affiliations:** 1Hellenic Cord Blood Bank, Biomedical Research Foundation Academy of Athens, 115 27 Athens, Greece; 2Histocompatibiliy and Immunogenetics Lab, Biomedical Research Foundation Academy of Athens, 115 27 Athens, Greece

**Keywords:** mesenchymal stromal cells, human leukocyte antigens, interferon-γ, autoimmune disorders, advanced therapy medicinal products

## Abstract

Mesenchymal stromal cells (MSCs) exert their immunoregulatory properties after licensing by inflammatory signaling cues, e.g., interferon (IFN)-γ. However, MSCs licensing by IFN-γ may result in increased expression of human leukocyte antigen (HLA) class II, which is related to rapid cell elimination, impairment of their immunosuppressive properties, and patient sensitization. The aim of this study was to evaluate the impact of IFN-γ on mediated immunoregulation and HLA class II expression. In this study, Wharton’s jelly (WJ) MSCs were isolated from human umbilical cords. Well-defined WJ-MSCs were submitted to IFN-γ exposure, and after 96 h, evaluation of biomolecule secretion and HLA class II expression was performed. Typing of HLA alleles using a next-generation sequencing (NGS) platform was performed. IFN-γ-primed WJ-MSCs secreted a high amount of immunoregulatory biomolecules, while elevated expression of HLA-DRB1 was observed. Analyses the NGS results showed the possibility of WJ-MSCs cluster formation based on their frequency of detected HLA alleles and immunoregulatory potential. Taking into consideration that IFN-γ-primed WJ-MSCs express HLA class II alleles, it is suggested that the HLA histocompatibility between allogeneic donor and recipient should be strongly considered to acquire the most beneficial outcome for the MSCs therapeutic strategy.

## 1. Introduction

MSCs have attracted the attention of scientific society as a therapeutic stem-cell-based approach for the proper administration of inflammatory, autoimmune, and other immune-related diseases, which have remarkably risen over the last decade [1]. MSCs represent a multipotent cellular population of mesodermal origin, equipped with great regenerative and immunoregulatory abilities [2,3]. Notably, the application of either autologous or allogeneic MSCs in phase I and II clinical trials has provided significant evidence for their safety and tolerability in humans [4,5]. Recently, MSCs have gained FDA approval for the management of steroid-refractory acute graft-versus-host disease (SR-aGVHD) in pediatric patients [6].

Nowadays, MSCs can be efficiently isolated from a variety of different sources from the human body, including the bone marrow (BM), adipose tissue (AT), stromal vascular fraction (SVF), Wharton’s jelly (WJ) tissue of the umbilical cord (UC), amniotic fluid (AF), placenta (P), dental teeth (DT), etc. [7,8]. Among them, BM represents the primary source for MSCs harvesting, since Alexander Fridenstein confirmed their presence and proposed their contribution in BM stroma formation [9]. Based on their origin, MSCs can be distinguished into, fetal-derived stem cells, e.g., AF-MSCs and WJ-MSCs, and adult-derived stem cells, e.g., BM-MSCs, AT-MSCs, and SVF-MSCs. MSCs derived from different sources exhibit distinct biological characteristics, including primarily the proliferation potential, stemness capacity, differentiation ability, and immunomodulatory action [10]. In 2006, the International Society for Cell and Gene Therapy (ISCT) proposed specific criteria for the proper characterization of MSCs, which have been recently updated [11,12]. Specifically, for defining MSCs the ISCT has proposed the following: (1) the plastic adherence ability; (2) multi-directional differentiation to mesodermal committed cell lineages (“osteocytes”, “chondrocytes” and “adipocytes”); (3) specific CD markers expression, including the positive expression of CD73, CD90, and CD105 (>90%) and the negative expression of CD34, CD45, CD11b, and HLA-DR (<3%) [11,12]. Beyond their regenerative potential, nowadays, well-defined MSCs are applied in several immune-related disorders, utilizing their beneficial immunoregulatory properties.

The licensing of MSCs by inflammatory signaling cues is considered a fundamental step in presenting their immunoregulatory abilities [13]. Interestingly, IFN-γ is considered the most potent stimulator upon binding to the IFNG receptor (IFNGR), which results in activation of the Janus kinase (JAK)–signal transducer and activator of transcription (STAT) intracellular pathway. This in turn leads to interferon regulatory factor 1 (IRF1) activation and expression of immunoregulatory membrane-bound and soluble-secreted biomolecules, including programmed death ligand (PDL)-1; indoleamine-2,3-dioxygenase (IDO); nitric oxide (NO); prostaglandin E2 (PGE2); galectins; anti-inflammatory cytokines, e.g., interleukin (IL)-1 receptor antagonist (RA), IL10, and IL13; and growth factors with known immunomodulatory action such as transforming growth factor-β1 (TGF-β1), fibroblast growth factor (FGF), hepatocyte growth factor (HGF), platelet derived growth factor (PDGF), and vascular endothelial growth factor (VEGF) [13]. In addition, IFN-γ is implicated in the signaling pathway of mitogen-activated protein 3 (MAP3) kinase, resulting in phosphorylation and nuclear translocation of p38 MAP kinase. This is further associated with gene expression of the aforementioned molecules, thus contributing to the adaptation of MSCs’ immunoregulatory role [13]. Once antigenic epitope presentation is initiated by the professional antigen-presenting cells (APCs), recognition by the cells of adaptive immunity, e.g., CD4+ or CD8+ T cells, follows, resulting in the secretion of high amounts of pro-inflammatory cytokines such as IFN-γ, TNF-α, and IL-1β, chemokines originating from the damaged cells and tissues, such as CCL2, CXCL10, and CCL19, and pro-inflammatory cytokines, e.g., TNF-α, IL-1βIL-3, and [14]. In the context of immune response initiation, MSCs receive these signals, increase the expression of several surface molecules, including CXCR1, CXCR2, CXCR3, CXCR4, VCAM, ICAM, etc., migrate to the damaged tissue, and upon stimulation by the inflammatory cues, mainly IFN-γ, increase the expression of HLA-class II, thus contributing to the antigen presentation process [15]. On the other hand, when the immune responses are persistent for a long time, higher levels of IFN-γ and other inflammatory biomolecules are produced by the majority of the immune cells. This event could lead to the initiation of severe pathogenetic damage to host tissues, which contributes to the accumulation of different immune cells at the inflamed area, including Th17 cells, which eventually can lead to the initiation of immune-related disorders [16,17]. To avoid this unfavored situation, and as part of the homeostatic action of MSCs, these stem cells can acquire an immunoregulatory phenotype to potentially tolerate the persistently overactivated immune responses [18]. For this purpose, MSCs are widely used as potent immunosuppressive agents in severe conditioned disorders, including the GVHD, induced after hematopoietic stem cells (HSCs) or solid organ transplantation, or to halt cytokine release syndrome (CRS) in COVID-19 patients [19]. Interestingly, recipients after haploidentical HSCs transplantation are characterized by a high possibility of acute GVHD (aGVHD) occurrence (30–63%) within the first 100 days of transplantation [20]. AGVHD is also induced by treatment-related tissue damage, where the donor’s T cells recognize and attack the recipients’ HLA-mismatched cells [21]. In addition, activated donor T cells migrate to other organs, causing extensive damage and the production of high levels of inflammatory mediators, a process which resembles several autoimmune disorders. Similar aGVHD rates have also been reported in solid organ transplantation [22]. To reverse the adverse events of aGVHD, steroid treatments such as prednisone or methylprednisolone are usually administered; however, a number of patients are less responsive to this therapeutic strategy, resulting in the development of a steroid-resistant (SR) condition [21]. Given that MSCs exert strong immunoregulatory potential, this stem cell therapy may be prove life-saving for patients with SR-aGVHD. Interestingly, results from phase I/II clinical trials have shown the beneficial effects of third-party MSCs, which are associated with no-rejection episodes and overall long-term patient survival [23,24,25,26]. Overall, MSCs appear to play a beneficial role in suppressing overactive immune responses. However, enhanced HLA class II expression in IFN-γ-primed MSCs can also be observed, which raises the following concern. Could the immunoregulatory properties of MSCs be compromised by the expression of HLA class II upon IFN-γ licensing? Upregulation of HLA class II expression in IFN-γ-primed MSCs can be further related to antigenicity acquisition, which could possibly elicit a host immune response against them and rapid clearance of the infused stem cells. In addition, there is a possibility of donor-specific antibody (DSA) formation against the infused third-party MSCs, leading to patients’ sensitization [27].

Considering the clinical utility of MSCs in immune-related disorders as potent immunosuppressive agents, the aim of this study was to insightfully explore the induction HLA class II expression and the immunoregulatory properties of IFN-γ-primed MSCs. The acquired data may offer additional insights into the role of MSCs as “sensors” and “switchers” implicated in both innate and adaptive immune responses. Interestingly, the acquired knowledge can be further utilized for the greater understanding of the MSCs’ immunobiology, which can be used as a basis for the development of “off-the-self” stem cell therapies to combat immune-related disorders.

## 2. Results

### 2.1. Characterization of Well-Defined Wj-Mscs

WJ-MSCs were successfully isolated from fresh hUC, characterized by spindle-shaped morphology under in vitro conditions. Moreover, the WJ-MSCs retained their fibroblastic-like morphology until they reached P4 (Figure 1). To test if MSCs fulfilled the minimum criteria outlined by the ISCT to be considered as well-defined stem cells, trilineage differentiation and immunophenotypic analysis were performed (Figure 1). WJ-MSCs successfully differentiated to “osteocytes”, “adipocytes”, and “chondrocytes”, as confirmed by alizarin red S, oil red O, and toluidine blue stains, respectively (Figure 1). WJ-MSCs exhibited positive expression of typical markers (CD73, CD90, and CD105) and negative expression of CD11b, CD34, and CD45 (Figure 1 and Appendix A). Specifically, the expression of WJ-MSCs at P4 for the classical CD markers, including the CD73, CD90, and CD105, was 96.7 ± 1.6%, 96.3 ± 1.9%, and 96 ± 1.6%, respectively, while for CD45, CD34, HLA-DR, CD15, CD3, CD19, CD31, and CD11b it was 1.1 ± 0.5%, 0.6 ± 0.3%, 1.1 ± 0.5%, 1.2 ± 0.6%, 0.6 ± 0.2%, 0.7 ± 0.3%, 0.5 ± 0.2%, and 0.6 ± 0.3%, respectively (Appendix A). WJ-MSCs at P4 also presented intermediate expression for CD49a (63.9 ± 9.8%) and CD44 (67.4 ± 8.3%). In addition, total cell number, CDT, and CPD were determined from P1 to P4. Specifically, the total numbers of MSCs for P1, P2, P3, and P4 were 2.1 × 10^5^ ± 3.9 × 10^5^, 1.9 × 10^6^ ± 3.3 × 10^6^, 5.8 × 10^6^ ± 1.2 × 10^6^, and 15.2 × 10^6^ ± 2.4 × 10^6^, respectively. The CDTs and CPDs of MSCs were 124.5 ± 14.4 h, 34.7 ± 5.2 h, 30.5 ± 8.3 h, 3.1 ± 0.3, 4.7 ± 0.4, and 6.1 ± 0.3 for P2, P3, and P4, respectively (Figure 1). The above results suggested that the MSCs were considered as well-defined stem cells based on the ISCT criteria, and can be used for the next series of experimental procedures.

### 2.2. Impact of IFN-γ Priming on WJ-MSCs Characteristics

Next, INF-γ was used as a stimulatory cue to activate the WJ-MSCs. Microscopically examination of IFN-γ-primed WJ-MSCs exhibited an increased number of intracellular vesicles compared to non-primed WJ-MSCs (Figure 2 and Appendix A). Total cell number and viability of non-primed and IFN-γ-primed WJ-MSCs were examined. Interestingly, the number of non-primed was 15.2 ± 1.8 × 10^6^ and for IFN-γ-primed WJ-MSCs it was 15.4 ± 1.7 × 10^6^ cells (Figure 2). The viability of non-primed WJ-MSCs was 92.2 ± 2.4% and of IFN-γ-primed WJ-MSCs it was 93.5 ± 3.1% (Figure 2), using the trypan blue method. Further verification of IFN-γ-primed WJ-MSCs viability was performed using the 7AAD stain in flow cytometry. Both methods presented similar viability results regarding the IFN-γ-primed WJ-MSCs viability (>92% either with trypan blue or 7AAD stains, Appendix A). To further assess the effect of IFN-γ on MSCs’ metabolic activity, total ATP and the ratio of ADP/ATP were determined. The total ATP of non-primed and IFN-γ-primed WJ-MSCs was 69.1 ± 11.8 μM and 70.4 ± 12.6 μM, respectively, whereas, of the positive control group the total was 1.8 ± 0.6 μM (Figure 2). The ADP/ATP ratio of the non-primed and IFN-γ-primed WJ-MSCs was 0.1 ± 0.05 for both groups, and 0.6 ± 0.1 for the positive control group (Figure 2). Between non-primed and IFN-γ-primed WJ-MSCs, no statistically significant differences were observed. The only statistically significant differences observed between the non-primed and IFN-γ-primed WJ-MSCs were against the positive control group regarding the total ATP concentration (*p* < 0.01) and ADP/ ATP ratio (*p* < 0.01). Immunophenotypic analysis indicated the upregulation of HLA-DR, -DQ, -DP, and CD10 expression (63.9 ± 9.8%, 62.1 ± 4.1%, 61.4 ± 9.8%, and 91.4 ± 4.2%, respectively) in IFN-γ-primed WJ-MSCs, whereas no altered expression levels regarding CD340 and CD49a were observed between IFN-γ-primed and non-primed WJ-MSCs (Figure 2 and Appendix A). Additionally, no alterations in levels of CD73, CD90, CD105, CD34, CD45, CD29, CD15, CD44, CD3, CD19, CD31, CD11b, CD340, CD80, and CD86 were observed between non-primed and IFN-γ-primed WJ-MSCs (Appendix A and Appendix A). Statistically significant differences (*p* < 0.001) were observed between non-primed and IFN-γ-primed WJ-MSCs regarding the expression of HLA-DR, -DQ, -DP, and CD10 (Figure 2 and Appendix A). Moreover, higher expression of p38 MAP kinase was observed in IFN-γ-primed WJ-MSCs compared to non-primed WJ-MSCs, as indicated by the results of indirect immunofluorescence (Figure 2). In regard to MLR, in both direct and indirect approaches IFN-γ-primed and non-primed WJ-MSCs were able to decrease the proliferation of CB-T cells (Figure 2). Of particular interest, under direct contact conditions non-primed WJ-MSCs resulted in a 29% reduction in T cell number, whereas IFN-γ-primed WJ-MSCs achieved a 66% reduction (Figure 2 and Appendix A). Under indirect contact, the reduction in T cell number was 27% and 50% for non-primed and IFN-γ-primed WJ-MSCs (Figure 2 and Appendix A). Statistically significant differences in MLR were observed between IFN-γ-primed and non-primed WJ-MSCs in direct (*p* < 0.001) and indirect (*p* < 0.001) approaches. Gene expression analysis showed the positive expression of *REX1*, *OCT4*, *NANOG*, *SOX9*, *KLF4*, and *GAPDH* in non-primed WJ-MSCs (Figure 2 and Appendix A). On the other hand, IFN-γ-primed WJ-MSCs were characterized by lower expression of *REX1*, while no significant alterations in gene expression of *OCT4*, *NANOG*, *SOX9*, and *KLF4* were observed (Figure 2 and Appendix A). The above results were further confirmed by real-time PCR, where a statistically significant difference in *REX1* expression between IFN-γ-primed and non-primed WJ-MSCs *(p* < 0.001) was indicated.

### 2.3. Biomolecules Quantification

IFN-γ-primed WJ-MSCs were characterized by an increase in secreted biomolecules compared to non-primed WJ-MSCs (Figure 3). Specifically, the content of IL-1Ra, IL-6, IL-10, and IL-13 of IFN-γ-primed WJ-MSCs was 811 ± 193 pg/mL, 86 ± 19 pg/mL, 238 ± 64 pg/mL, and 150 ± 39 pg/mL, respectively. The content of the same cytokines for the non-primed WJ-MSCs were 86 ± 25 pg/mL, 32 ± 9 pg/mL, 43 ± 18 pg/mL, and 41 ± 15 pg/mL, respectively (Figure 3). Statistically significant differences were observed between the IFN-γ-primed and non-primed WJ-MSCs regarding the aforementioned determined cytokines (*p* < 0.001). In a similar manner, the levels of TGF-β1, FGF, PDGF, VEGFA, HGF, NO, and IDO of the IFN-γ-primed WJ-MSCs were 873 ± 132 pg/mL, 887 ± 142 pg/mL, 949 ± 156 pg/mL, 1026 ± 182 pg/mL, 880 ± 184 pg/mL, 63 ± 16 μM, and 1095 ± 258 pg/mL, respectively, while for the non-primed WJ-MSCs they were 509 ± 111 pg/mL, 448 ± 73 pg/mL, 390 ± 105 pg/mL, 491 ± 119 pg/mL, 405 ± 158 pg/mL, 12 ± 5 μM, and 328 ± 136 pg/mL (Figure 3). Statistically significant differences were observed between IFN-γ-primed and non-primed WJ-MSCs regarding the above-mentioned biomolecules (*p* < 0.001). In addition, uMAP analysis showed that 15 out of 20 MSCs samples (75%) were characterized by increased immunoregulatory biomolecule secretion; however, 25% of the MSCs samples were characterized by lower immunoregulatory biomolecule secretion, which was comparable to non-primed WJ-MSCs. A functional protein association network showed that IL-6 and TGF-β1 were the main orchestrators of anti-inflammatory responses, and that they actively interacted and possibly regulated the production of the immunoregulatory biomolecules secreted by the IFN-γ-primed-WJ-MSCs.

### 2.4. Analysis of HLA Class I and II Alleles in WJ-MSCs

High-resolution HLA typing using the NGS approach was performed to determine the most frequent HLA class I and II alleles presented by the WJ-MSCs. Regarding HLA-A, the most frequent HLA alleles were A*02:01:01 (23%), A*24:02:01 (18%), A*03:01:01 (15%), and A*01:01:01 (13%), thus representing 69% of the total HLA-A alleles (Figure 4A). For HLA-B, the most frequent HLA alleles were B*51:01:01 (18%), *35:01:01 (10%), *35:03:01 (10%), *18:01:01 (8%), and *44:02:01 (8%), and for HLA-C, C*04:01:01 (28%), *12:03:01 (20%), *06:02:01 (8%), and *15:02:01 (8%) were most frequent, representing 54% and 64% of the total HLA-B and C alleles, respectively (Figure 4). In regard to HLA class II, the most frequent alleles for HLA-DRB1 were DRB1*11:04:01 (15%), *11:01:01 (13%), *14:54:01 (10%), *13:01:01 (8%), and *01:01:01 (8%), for HLA-DQB1 they were DQB1*03:01:01 (40%), *05:03:01 (15%), *05:01:01 (13%), and *05:02:01 (8%), and for HLA-DPB1 they were DPB1*04:01:01 (50%), *04:02:01 (15%), *02:01:02 (10%), and *03:01:01 (8%), representing 54%, 76%, and 83% for HLA-DRB1, -DQB1, and -DPB1 alleles, respectively (Figure 4). The neighbor-joining trees for HLA class I and II depicted the genetic distance between the HLA alleles. In addition, similar antigenic epitopes represented by different HLA alleles were found within the same phylogenetic branch (Figure 4 and Appendix A). Based on the HLA class I and II allele frequencies and biomolecule secretory profiles, clustering of the different MSC samples was performed, as shown in Figure 4D. Indeed, three major MSCs clusters were formed, with the first cluster including primarily MSC batches with the greatest immunoregulatory profile in terms of secretion and most frequent HLA class I and II alleles (Appendix A). Moreover, nucleotide comparison of core exons in HLA class I and II molecules depicted significant alterations between the different HLA alleles (Figure 5). Specifically, for HLA class I, including HLA-A, -B, and -C alleles, multiple nucleotide alterations were observed in exon 2 position (pos) 74G-342C and exon 3 pos 343G-619G (Appendix A). Regarding HLA class II, nucleotide alterations were also observed in HLA-DRB1 pos 100C-369A, in HLA-DQB1 pos 109G-378A, and in HLA-DPB1 pos 100G-383A (Appendix A). These nucleotide alterations result in the different three-dimensional structure formations of the different HLA alleles. However, when mismatches between donor and recipient exist, these nucleotide mismatches are responsible for the initiation of specific immune responses towards the foreign HLA alleles, which is further associated with the production of DSAs.

## 3. Discussion

The clinical utility of MSCs was demonstrated 20 years ago when they were used to treat severe steroid-resistant GVHD in patients who had undergone transplantation [25]. Since then, over 900 clinical trials (www.clinicaltrials.gov) have been conducted, involving both autologous and allogeneic third-party MSCs. A great number of those clinical trials were focused on the proper administration of GVHD [28], autoimmune disorders [29], and osteoarticular damaged tissue [30], while recently WJ-MSCs were applied in phase I/II trials for halting CRS mediated by SARS-CoV-2 pathology [31]. The Food and Drug Administration (FDA) approved in 2024, MSCs application in pediatric patients suffering from steroid-resistant GVHD [6]. Several preclinical and clinical studies have prompted the idea to re-examine the immune-privileged status of MSCs and reconsider the concept of using them universally as “off-the-self” cellular therapy [32,33]. Whilst most studies have clearly shown the beneficial effect of MSCs in terms of immunosuppression, there is evidence in the literature indicating that HLA-mismatched MSCs can elicit host immune responses [32,33]. Previous studies have demonstrated that MSCs can modify their characteristics, notably exhibiting increased expression of HLA class II molecules when activated by an inflammatory stimuli, in particular by IFN-γ [13,34,35]. IFN-γ is considered the master regulator for the secretion of MSCs’ immunoregulatory biomolecules [36,37]. Accordingly, in this study, we investigated the impact of IFN-γ licensing on MSCs’ immunobiology by exploring the acquired immunoregulatory phenotype and the altered HLA class II expression. The WJ-MSCs used in this study were characterized as well-defined stem cells based on the criteria that have been outlined by the ISCT. Briefly, WJ-MSCs presented a fibroblastic spindle-shaped morphology, followed by successful trilineage differentiation (“osteocytes”, “adipocytes”, and “chondrocytes”) and typical CD markers expression (CD73, CD90, and CD105 > 95%, and CD15, CD34, CD45, and HLA-DR < 3%). In addition to the markers above, WJ-MSCs showed moderate to high expression of CD29, CD44, and CD49α. Following IFN-γ exposure, primed WJ-MSCs were characterized by an increased number of intracellular vesicles when compared to non-primed WJ-MSCs; however, no other morphological alterations, in terms of size or shape, were observed. Similar observations in IFN-γ-primed WJ-MSCs have been reported by other groups in the past, where primary senescent-like characteristics have been found, including the accumulation of reactive oxygen species (ROS) after IFN-γ licensing [38]. However, in our study, no cytotoxicity of WJ-MSCs was evident after 96 h exposure to IFN-γ, as confirmed by the determination of cell number, viability, ATP, and ADP/ATP ratio. No statistically significant differences regarding the characteristics mentioned above were found between IFN-γ-primed and non-primed WJ-MSCs. This suggested that the cytotoxicity reported by Yang ZX et al. [38] may be attributed to variations regarding the different origins of MSCs (e.g., bone marrow-derived MSCs) and also to the different applied experimental conditions compared to this study. IFN-γ-primed WJ-MSCs were characterized by statistically significant upregulation of HLA-DR, HLA-DQ, HLA-DP, and CD10 expression compared to non-primed WJ-MSCs, while CD340 and CD49a did not present any such alteration. CD340 can be considered as an advanced marker for MSCs characterization, mostly related to the proliferation potential of MSCs, while CD49a is the α1 integrin subunit, which is related to the migration and homing abilities of MSCs [39]. No alterations in either characteristic were observed in MSCs after IFN-γ licensing. Regarding the increased expression of HLA class II (including HLA-DR, -DQ, and -DP), it has already been shown that IFN-γ mediates phosphorylation of STAT1, causes the strong activation of Class II major histocompatibility complex transactivator (CIITA), and eventually the transcription of all HLA class II genes (e.g., DR, DQ, DP) [40]. This is the primary reason why, in our study, production of membrane-bound HLA class II molecules was observed upon IFN-γ licensing of MSCs. The increased expression of membrane-bound HLA-DR seems to be in accordance with the studies of Le Blanc et al. [34], van Megen et al. [35], and Kuci et al. [41]. Moreover, it has been shown that besides the upregulation of HLA class II expression in IFN-γ-primed MSCs, no presence of co-stimulatory molecules, including CD20L, CD80 (B7-1), and CD86, was evident in MSCs [42,43,44]. This fact was also confirmed in our study, where no increases in CD80 and CD86 expression were noticed in IFN-γ-primed WJ-MSCs. In regard to increased expression of CD10, Kouroupis et al. [45] presented evidence for a potential link with the enhanced immunosuppressive properties exerted by the IFN-γ-licensed MSCs. Taking into account the above, IFN-γ exposure may lead to upregulation of CD10 expression, thus further contributing to the immunoregulatory phenotype of WJ-MSCs. However, to properly clarify the association between IFN-γ and increased expression of CD10 by primed MSCs, further experimental procedures are required.

Knowing that IFN-γ favors the immunoregulatory phenotype of MSCs through the activation of various intracellular signaling pathways, the next step of this study was the evaluation of altered p38 MAP kinase expression in primed MSCs. Indeed, IFN-γ-primed WJ-MSCs are characterized by high intranuclear localization of p38 MAP kinase, an event which is further associated with anti-inflammatory cytokine gene transcription. It is well known that upon exposure of MSCs to IFN-γ, activation of MAP kinase, including primarily p38 MAP kinase, is performed, a process which is mediated by the phosphorylated JAK1/2 and STATs [46,47,48]. As part of a greater understanding of primed MSCs’ immunobiology, gene expression levels of *REX1*, *OCT4*, *NANOG*, and *SOX9* were evaluated. REX1 belongs to the YY1 sub-family of transcription factors, which can act as a transcription regulator for a great number of genes, influencing primarily the stemness, cell growth, and protein secretion [49,50]. In stem cells, REX1 is considered alongside OCT4 and NANOG as a key-specific marker for stemness preservation. One of the primary functions of REX1 is the inhibition of the JAK/STAT pathway, thus acting competitively with p38 MAP kinase [49,50]. In this study, non-primed WJ-MSCs were characterized by high expression of *REX1* and low detection of p38 MAP kinase (mainly cytoplasmic). On the other hand, in IFN-γ-primed WJ-MSCs, intranuclear translocation of p38 MAP kinase and a significant decrease in *REX1* expression were observed. No differences in the gene expression levels of *OCT4*, *KLF4*, *NANOG*, and *SOX9* were observed, reflecting the preservation of stemness potential in MSCs after IFN-γ stimulation. To further explore the immunoregulatory action of primed WJ-MSCs, MLR experiments were conducted. These experiments proved that primed WJ-MSCs effectively induced the decrease in T cell numbers, thus confirming that IFN-γ plays a crucial role in immunoregulatory phenotype adaptation by MSCs, where the interplay between *REX1* and p38 MAP kinase is important.

Considering that secreted immunoregulatory biomolecules attract the interest of scientific society in terms of using them as “off-the-self” therapy, quantification of those proteins was performed. IFN-γ-primed WJ-MSCs secreted a great amount of immunoregulatory biomolecules, including cytokines such as IL-1RA, IL-10, and IL-13, growth factors such as TGF-β1, FGF, PDGF, HGF, and VEGF, and other biomolecules including IDO and NO. Beyond the above cytokines, elevated levels of IL-6 were evidenced in the IFN-γ-primed WJ-MSCs. Among them, HGF had the greatest expression, followed by VEGFA, PDGF, and IDO. The protein interactome revealed that TGF-β, IL-6, and IL-13 are key protein mediators, and that they actively associate with other biomolecules. Indeed, it has been shown that IL-6 exerts a pleotropic action, regulating possibly the balance between pro- and anti-inflammatory cytokine production [51]. Moreover, supporting evidence is provided by the fact that IL-6-deficient mice failed to produce proper anti-inflammatory responses, an event which contributes to the initiation of autoimmune disorders [51,52,53]. IL-6, produced by IFN-γ-primed WJ-MSCs, can also act in an autocrine manner, influencing the production of other anti-inflammatory cytokines and favoring Th2 shifting. In line with this, IL-13 favors going beyond Th2 shifting to M2 phenotype macrophage polarization. In addition, IL-13 can reduce the production levels of TNF-α, IL-1β, and IL-3 through the blockage of nuclear factor-kappa B (NF-kB) and c-Jun terminal kinase (JNKs), contributing to the immunomodulation of the overactivated immune responses [54]. Recently, it has been shown that TGF-β1, independently of the suppressor of mothers against decapentaplegic (SMADs) signaling pathway, is associated with the activation of p38 MAP kinase [55]. Indeed, activation of the TGF-β1 type 1 receptor promotes the phosphorylation of SMAD2 and SMAD3, resulting in a plethora of biological events, e.g., cell growth arrest and apoptosis, as part of its canonical signaling pathway [55,56]. However, it has been shown that TGF-β1 can mediate the activation of other significant signaling proteins, such as several MAP kinases, extracellular signal-regulated kinases (ERKs), JNKs, and p38 MAP kinase, by engaging a currently unknown SMAD-independent signaling pathway [55]. Knowing that TGF-β1, which is secreted by the IFN-γ-primed WJ-MSCs, has an autocrine effect, this growth factor can influence diverse biological consequences, contributing to elevated levels of p38 MAP kinase, which are finally associated with the adaptation of immunoregulatory phenotype by WJ-MSCs. It is noteworthy to mention that only 25% of IFN-γ-primed WJ-MSCs are characterized by low secretion of immunoregulatory biomolecules. To further explain this phenomenon, IFNGR1 sequencing could be a future target of evaluation to shed light on whether specific structural mutations impair the binding affinity of IFN-γ, resulting in poor MSC response. However, this is beyond the scope of this study and could be a future direction to better understand MSCs’ immunobiology.

Based on the current literature and from the results provided herein, non-primed WJ-MSCs did not express membrane-bound HLA class II molecules [57]. On the other hand, IFN-γ can efficiently cause the priming of WJ-MSCs and the expression of membrane-bound HLA class II molecules [36]. Recent evidence has shown that MSCs are broadly applied as third-party allogeneic advanced therapeutic medicinal products (ATMPs) to a great number of immune-related diseases, including autoimmune disorders [1,28,29,31]. In the majority of those cases, no HLA matching between donor and recipient was required, which may raise the question of whether allogeneic MSCs are capable of inducing DSAs formation, resulting in a rapid clearance of cells which is accompanied by the subsequent sensitization of the recipient. Trying to answer this question, high-resolution HLA typing showed that the WJ-MSCs characterized by the most frequent HLA alleles are found in the Greek population [58]. In addition, neighbor-joining trees for HLA class I and II alleles showed the existence of several branches, with HLAs that belong to the same branch characterized by similar binding affinities for specific antigenic epitopes. By combining the secretory profile of MSCs and their HLA class I and II molecules, clustering of the WJ-MSCs was achieved. Cluster analysis indicated that MSCs can be classified according to their secretory profiles and HLA allele frequencies into groups reflecting high, moderate, and low responses. For a future perspective, this could lead to the establishment of a cell biobank with well-defined MSC lines which are readily accessible upon demand. Also, this may further explore the possibility of using MSCs precisely selected for a specific recipient (based on donor–recipient histocompatibility), which would elicit the most beneficial outcome for the patient. A notable example is the HLA alleles associations with disease susceptibility in various immune-related disorders, including multiple sclerosis (MS) [59,60,61,62]. For instance, high-risk HLA alleles, including HLA-DRB1*15:01, HLA-DQB1*06:02, and HLA-DQB1*01:02, have been reported in Caucasian patients suffering from MS [63]. In this way, myelin basic protein (MBP) and myelin oligodendrocyte glycoprotein (MOG) epitopes are strongly expressed by the aforementioned HLA alleles, triggering the CD4+ T cell and CD19+ B cell responses and contributing to disease pathogenesis [64,65,66]. For this purpose, it is possible that the administration of precisely selected third-party MSCs, bearing HLA alleles such as HLA-DRB1*01:01, HLA-DRB1*11:01, HLA-DRB1*07:01, and HLA-DQB1*06:02 with known protective action against MS, may exert better immunoregulatory action against the adaptive immunity cells in those patients [67,68].

A meta-analytic study conducted by Bezstaroti et al. [69] showed the possibility of DSA formation against non-HLA-matched infused MSCs in kidney transplanted patients. It was indicated that to avoid repeated HLA mismatches, high-resolution HLA typing at three fields (all exons sequencing) should be performed. To address this, HLA typing with NGS was performed in our study, revealing specific nucleotide alterations in exons 2 and 3 for HLA class I and exon 2 for HLA class II alleles. Interestingly, exons 2 and 3 in HLA class I and exon 2 in HLA class II contribute to the formation of the antigen-binding groove. In this way, alterations in these positions can be further related to the induction of DSA formation when HLA mismatches exist between donor and recipient, resulting in the rapid clearance of the infused MSCs. Moreover, this event can also be associated with impaired immunoregulatory action by MSCs due to their clearance by host defense. Apart from the above, DSA formation is actively related to patients’ sensitization, which represents a negative event especially for severely conditioned patients, such as kidney transplant recipients or those who belong to organ waiting lists. Sensitization of the recipient represents a serious condition which hampers the possibility of finding a histocompatible donor, whilst it is also responsible for the higher possibility of aGVHD occurrence after transplantation. Moreover, recently it has been shown that nucleotide alterations in HLA alleles between donor and recipient beyond exon 3 for HLA class I and exon 2 of HLA class II, known as hidden patterns, may be subject to host immune recognition and production of DSAs [70]. Therefore, in the case of severely conditioned patients experiencing an immune-related disorder, where MSCs could be applied as a therapeutic strategy, the HLA histocompatibility of MSCs using the NGS approach should be performed and importantly taken into account by the physician for proper MSC line selection.

In this study, it was clearly shown that IFN-γ-primed MSCs secrete key immunomodulatory biomolecules, contributing to the toleration of the overstimulated immune responses; however, a significant upregulation of HLA class II expression was also noted. Since HLA class II may trigger the development of DSAs in the recipient, this finding may be further correlated with an impaired immune tolerance exerted by MSCs. To further elucidate the possibility of functional consequences of HLA class II upregulation, which are expressed by MSCs after infusion in patients suffering from immune-related disorders, in vivo studies with specific animal models are required. This fact represents the main limitation of the present work, which will be explored in the near future taking into account the results presented herein. Specifically, autoimmune humanized animal models should be employed to assess the functional relevance of HLA class II in DSA formation. Overall, the obtained results provide new insights to better understand the immunobiology of MSCs, which can be potentially utilized as an alternative therapeutic strategy to combat human immune-related disorders.

## 4. Materials and Methods

### 4.1. Ethical Statement

The human umbilical cords used for the isolation of MSCs were accompanied by informed consent regarding their research use, signed by the mothers a few days before the delivery. The informed consent was in accordance with the Declaration of Helsinki and conformed to the ethical standards of the Greek National Ethical Committee. The overall study has been approved by the Biomedical Research Foundation Academy of Athens (BRFAA) Bioethics Committee (Reference No. 1754, 21 January 2021).

### 4.2. Isolation and Expansion of WJ-MSCs

MSCs were isolated from the WJ tissue of human umbilical cords (hUCs), which were delivered to the Hellenic cord blood bank (HCBB) within 24 h after gestation. In total, 50 full-term (with gestational weeks 38–40) hUCs derived either from normal or caesarian deliveries were used for the experiments performed in this study. After hUCs reception by the personnel of the HCBB, they were immediately processed or remained for a maximum of 48 h at 4 °C. Before the MSCs isolation, the hUCs were extensively washed with phosphate-buffer saline 1× (PBS 1×, Sigma-Aldrich, Darmstadt, Germany) to remove any blood clots from the entire cord. Then, using sterile surgical tools, the umbilical arteries and veins were isolated and removed, leaving only the Wharton’s jelly (WJ) tissue. The WJ tissue was dissected into small pieces (0.3 cm × 0.3 cm) and placed in a 6-well culture plate (Costar, Corning, Life, Canton, MA, USA) for 18 days (Appendix A). Complete culture medium consisted of a-Minimum Essentials Medium (MEM, Sigma-Aldrich, Darmstadt, Germany), supplemented with 20% fetal bovine serum (FBS, Sigma-Aldrich) and 1% penicillin–streptomycin (Sigma-Aldrich, Darmstadt, Germany), and 1% L-glutamine (Sigma-Aldrich, Darmstadt, Germany) was added (1 mL/well plate). The tissue cultures were incubated for a total period of 18 days at 37 °C and 5% CO_2_. After 18 days of culturing, the plates were examined under a light microscope and upon confluency of at least 3 of 6 wells, passage was performed.

To perform the MSCs passage, the culture medium was removed, followed by PBS 1× addition, which remained for 1–3 min maximum. The PBS 1× was completely removed, and 500 μL of trypsin solution (Trypsin-EDTA solution 0.05% *w*/*v*, Gibco, Thermo Fischer Scientific, Waltham, MA, USA) per well was added. The culture plates were placed in the incubator for 10 min at 37 °C. Detached cells were transferred into a 75 cm^2^ culture flask. The whole procedure was repeated until passage 4 (P4).

### 4.3. Growth Kinetics of WJ-MSCs

To further validate the WJ-MSCs characteristics (from P1 to P4), total cell number, cell doubling time (CDT), and cumulative population doubling (CPD) were determined. Initially, a number of WJ-MSCs ranging from 1.5–2 × 10^5^ were harvested from the 6-well-plates and transferred to 75 cm^2^ cell culture flasks (Costar, Corning Life, Canton, MA, USA), followed by passages at dedicated time points. Between each passage, MSCs number counting was performed using a Countess 3 Automated Cell Counter (ThermoFischer Scientific, Waltham, MA, USA). For counting performance, trypan blue in dilution 1:1 was used also (by applying 10 μL of MSCs yield to 10 μL of Trypan blue solution 1%), or if it was necessary, dilution of the harvested cells 1:5 or 1:10 was performed. The time required for P1 was 432 h, while for P2 to P4 the time required between each passage was 240 h.

The estimation of CDT was performed using the following equation:CDT = log10(*N*/*N0*) ÷ log10(2) × *T*

The estimation of PD was determined using the following equation:PD = log10(*N*/*N0*) ÷ log10(2)

The cumulative PD was obtained after the addition of the passage’s current PD to the calculated PD from the previous passage.

In the above equations, *N* was the number of cells at each passage, *N0* was the number of initially plated WJ-MSCs, and *T* was the culture duration in hrs.

### 4.4. Immunophenotypic Analysis of WJ-MSCs

Immunophenotypic analysis of WJ-MSCs (*n* = 30) at P4 was performed using a panel of 11 monoclonal antibodies (mAbs). Specifically, fluorescein (FITC)-labeled antibodies against CD90, CD45, CD29, CD31, and HLA-ABC, phycoerythrin (PE)-labeled mAbs against CD44, CD3, CD11b, and CD34, peridinin–chlorophyll–protein (PerCP)-labeled antibodies against CD19, CD105, and HLA-DR, and allophycocyanin (APC)-labeled antibodies against CD15 and CD73 were applied. All mAbs were provided by BD Biosciences (BD, Biosciences, Franklin Lakes, NJ, USA). The immunophenotypic analysis of WJ-MSCs was performed in FACSCanto II (BD, Biosciences, Franklin Lakes, NJ, USA), and all Abs used in the current protocol were purchased from BD Biosciences. WJ-MSCs P4 in a total number of 1.4 × 10^6^ cells were harvested, transferred to 7 polystyrene tubes (2 × 10^5^ cells/tube), and incubated with the above mAbs for 20 min in the dark place. Then, all tubes were centrifuged at 500× *g* for 6 min, the supernatant was discarded, followed by the addition of 1 mL of PBS 1× (Gibco, ThermoFischer Scientific, Waltham, USA). Finally, the tubes were loaded in the carousel, in a dedicated order, and immunophenotyping was performed. A total of 100.000 events were acquired for each parameter tested. Complete flow cytometric analysis was performed in FlowJo v10 (BD, Biosciences, Franklin Lakes, NJ, USA).

### 4.5. Trilineage Differentiation of WJ-MSCs

To confirm the multipotentiality of WJ-MSCs at P4 (*n* = 5), trilineage differentiation to “osteocytes”, “adipocytes”, and “chondrocytes” was performed. For this purpose, StemPro Osteogenesis, Adipogenesis, and Chondrogenesis kits (Thermo Fischer Scientific, Waltham, MA, USA) were used, following the manufacturer’s instructions. The successful differentiation of WJ-MSCs was confirmed with histological stains. Alizarin red S, oil red O, and toluidine blue (Sigma-Aldrich) were applied for the evaluation of calcium deposition formed by “osteocytes”, lipid droplets formed by “adipocytes”, and glycosaminoglycans (sGAGs) production by “chondrocytes”, respectively.

### 4.6. IFN-γ Priming of WJ-MSCs

WJ-MSCs P4 (*n* = 20) were initially seeded in a 25 cm^2^ tissue culture flask with 10 mL of complete culture medium. The next day, the WJ-MSCs were microscopically examined for the successful seeding and washed 3 times with PBS 1× (Sigma-Aldrich) to completely remove residual complete culture medium. Then, a stimulation medium consisting of 10 mL α-MEM (Sigma-Aldrich), 1% penicillin–streptomycin (Sigma-Aldrich), 1% L-glutamine (Sigma-Aldrich), and 100 ng/μL IFN-γ (Invitrogen, Carlsbad, CA, USA) was added to each sample. All WJ-MSCs samples were incubated at 37 °C and 5% CO_2_ for 96 h, with only a change in the stimulated medium contained 100 ng/μL ΙFN-γ. After 96 h, the IFN-γ-primed WJ-MSCs were used for the subsequent experiments.

### 4.7. Evaluation of IFN-γ-primed WJ-MSCs Properties

IFN-γ-primed WJ-MSCs P4 were further analyzed for their morphological features and immunophenotypic analyses and compared with non-primed WJ-MSCs as a reference control group. For this purpose, WJ-MSCs were also seeded in a 25 cm^2^ tissue culture flask with 10 mL of complete culture medium. After 96 h, the cells were detached using trypsin–EDTA 0.05% solution (Gibco, Thermo Fischer Scientific, Waltham, MA, USA) and the cells were counted using trypan blue in a Countess 3 automated cell counter, as previously described. For immunophenotypic analysis of IFN-γ-primed and non-primed WJ-MSCs, besides the CD markers described above, CD340-Percp-Cy5-5A, CD10-APC, CD80-PE, CD86-FITC, 7AAD (viability stain), HLA-DP-PE, and HLA-DQ-FITC were applied. All mAbs were provided by BD Biosciences (BD, Biosciences, Franklin Lakes, NJ, USA). The immunophenotypic analysis of non-primed and IFN-γ-primed WJ-MSCs was performed as described previously.

### 4.8. Estimation of Cell Proliferation Using the ATPAssay

The determination of the WJ-MSCs proliferation after IFN-γ priming was performed using a commercial ATP assay (MAK190, Sigma-Aldrich, Darmstadt, Germany). To estimate the cell proliferation, 1 × 10^5^ non-primed WJ-MSCs P4 (*n* = 10) or IFN-γ-primed WJ-MSCs P4 (*n* = 10) were seeded in 24-well plates (Costar, Corning Life, Canton, MA, USA). The following day, the cells from both groups were lysed with 100 μL ATP assay buffer and 20 μL of the cell lysates were transferred to 96-well plates, followed by the addition of the reaction buffer. The samples were incubated for 30 min at RT and then absorbance was measured in a Tristar 5 (Berthold Technologies GmbH & Co.KG, Bad Wildbad, Germany) photometer at 570 nm wavelength. Determination of the ATP concentration of each sample was performed by interpolation to a standard curve. The standard curve consisted of 0 (blank), 5, 10, 20, 50, 100, 150, and 200 nmol standards.

### 4.9. ADP/ATP Assay

To further validate the induced cell death by IFN-γ exposure to WJ-MSCs, an ADP/ATP assay was performed using a commercially available kit (MAK135, Sigma-Aldrich, Darmstadt, Germany). The assay was performed in non-primed WJ-MSCs (*n* = 10) and IFN-γ-primed WJ-MSCs (*n* = 10), following the manufacturer’s instructions. Briefly, the light intensity, which is specific to intracellular ATP concentration, is produced with the following reaction:ATP + D-Luciferin + O_2_ → oxyluciferin + AMP + PPi + CO_2_ + light

In the next step, the ADP is converted to ATP, which further reacts with D-luciferin. The second light intensity determines the total ADP and ATP concentration. The light intensity was measured using a luminometer (Lucy 1, Anthos, Luminoskan, Labsystems, Kaltenkirchen, Germany) and expressed as the number of relative light units (RLUs). The determination of ADP/ATP ratio is performed using the following formula:ADP/ATP ratio = (RLU C − RLU B) ÷ RLC A
where RLU A is the initial luminescence measurement after the addition of the ATP reagent, RLU B is the luminescence measurement after 10 min of incubation, and RLU C is the measurement of light intensity after the addition of ADP reagent. For the performance of the assay, non-primed WJ-MSCs (*n* = 10) and IFN-γ-primed WJ-MSCs (*n* = 10) were seeded at a density of 2 × 10^5^ cells in 24-well plates with 1 mL of complete culture medium. Finally, cell cultures were incubated for a total of 7 days in a humidified atmosphere and 5% CO_2_. WJ-MSCs with 10% *v*/*v* DMSO served as a positive control group. The determination of the ADP/ATP ratio was performed after 7 days.

### 4.10. Quantification of Biomolecules Produced from Primed WJ-MSCs

Biomolecules, including the anti-inflammatory cytokines and growth factors produced from IFN-γ-primed WJ-MSCs were quantified using commercial ELISA kits (OriGene Technologies, Maryland, USA). Specifically, in the presented study, the quantification of IL-1 receptor antagonist (RA), IL-6, Il-10, IL-13, TGF- β1, VEGFA, FGF-1, PDGF, HGF, IDO, and NO were performed following the manufacturer’s instructions. The quantification of biomolecules was performed on culture supernatants obtained from non-primed (*n* = 20) and IFN-γ-primed (*n* = 20) WJ-MSCs. Non-primed and IFN-γ-primed WJ-MSCs were washed 3 times with PBS 1× and cultured with a medium containing only α-ΜΕΜ supplemented with 1% penicillin–streptomycin (Sigma-Aldrich, Darmstadt, Germany) and 1% L-glutamine (Sigma-Aldrich, Darmstadt, Germany) for 48%. The supernatants were collected, and the performance of each enzyme-linked immunosorbent assay (ELISA) was evaluated. The final concentration of each biomolecule was estimated through interpolation to a standard curve.

### 4.11. Indirect Immunofluorescence for p38 MAP Kinase

Indirect immunofluorescence against p38 MAP kinase was performed in non-primed (*n* = 5) and IFN-γ-primed (*n* = 5) WJ-MSCs. For this assay, 1 × 10^4^ were seeded on culture slides (Sigma-Aldrich, Darmstadt, Germany). A total of 1 mL of complete culture medium was used for the non-primed WJ-MSCs, and 1 mL of complete culture medium supplemented with 100 ng/mL IFN-γ was used for the primed WJ-MSCs. The slides were observed microscopically for confluency achievement by WJ-MSCs and then indirect immunofluorescence was initiated. WJ-MSCs of both groups were fixed with 10% *v/v* neutral formalin buffer (Sigma-Aldrich, Darmstadt, Germany). Then, antigen retrieval and blocking of the WJ-MSCs were applied, followed by the addition of the primary monoclonal antibody against human p38 MAP kinase (1:1000, Catalog No 506123, Sigma-Aldrich, Darmstadt, Germany). The primary antibody was incubated overnight at 4 °C. The next day, the slides were rinsed in distilled H_2_O to remove non-specific antibody stain, followed by the addition of the secondary mouse-anti-human monoclonal antibody-FITC conjugated (1:100, Sigma-Aldrich, Darmstadt, Germany). The secondary antibody was incubated for 1 h in the dark at RT. DAPI stain was used for the observation of cell nuclei. The slides were rinsed with distilled H_2_O and finally were glycerol mounted and checked under a fluorescent microscope (Leica SP5 II). The images were acquired using a camera mounted in a microscope coupled with the software LAS Sute v2 (Leica, Microsystems, Wetzlar, Germany).

### 4.12. Mixed Lymphocyte Reaction

Mixed lymphocyte reaction (MLR) was performed to assess the immunosuppressive properties of IFN-γ-primed and non-primed WJ-MSCs through direct and indirect contact. Mononuclear cells were isolated from 5 mL of two different CBUs, using the Ficoll-Paque method (Cytiva Ficoll-Paque™ PLUS Media, Sigma-Aldrich, Darmstadt, Germany), as has been previously described [71]. An equal number of allogeneic CB-MNCs (obtained from a different donor), referred to as responders, were isolated, followed by magnetic separation of T cells using a Pan T Cell Isolation Kit (Miltenyi, Biotec, Bergisch Gladbach, Germany) to avoid activation of T cells. For the direct contact, 5 × 10^3^ CB-MNCs, referred as stimulators, were seeded in a 24-well plate (Costar, Corning Life, Canton, MA, USA) and treated with mitomycin 25 μg/mL (Sigma-Aldrich, Darmstadt, Germany) for 30 min at 37 °C. An equal number of allogeneic CB-T cells (obtained from a different donor), referred to as responders, was added to the culture. Then, 5 × 10^3^ of either IFN-γ-primed or non-primed WJ-MSCs were applied to further assess their effectiveness in halting lymphocyte proliferation. In each well, 1 mL of cultured medium was used. For the indirect contact, the same number of stimulator and responder cells were seeded on the insert (with a pore size 0.4 μm). IFN-γ-primed WJ-MSCs or non-primed WJ-MSCs were seeded on the bottom of the 24-well plate. The culture medium used in the above tasks consisted of a-MEM with 15% FBS, 1% P-S, and 1% L-glutamine. The total number of CB-T cells was counted with a Countess 3 automated cell counter (Thermo Fischer Scientific, Waltham, USA), as earlier described in this study. After 6 days (144 h), the CB-T cells were counted again, and the proliferation index was determined. In the above assays, as a positive control group, 5 × 10^3^ CB-T cells treated with 5 μg/mL phytohemagglutinin (PHA, Sigma-Aldrich, Darmstadt, Germany), and as a negative control, 5 × 10^3^ unstimulated CB-T cells, were applied.

### 4.13. Gene Expression Analysis

To evaluate the gene expression, total mRNA was initially isolated from non-primed (*n* = 3) and IFN-γ-primed WJ-MSCs (*n* = 3) using a Tri reagent (Sigma-Aldrich, Darmstadt, Germany) following the manufacturer’s instructions. Finally, the isolated mRNA was eluted in 30 μL of RNAse-free H_2_O. Quantification of mRNA concentration and purity from all samples was performed using a Nanodrop Lite spectrophotometer (Thermo Fischer Scientific) at A260/A280 ratio.

Then, reverse-transcription (RT) PCR was performed to obtain the cDNA. For this purpose, a engineered M-MLV reverse transcriptase basic kit (EnzyQuest, Heraklion, Crete, Greece) was used. Briefly, 1 μg of total RNA was added to a reaction buffer containing 2 μL oligo (T)18 50 μM, 1 μL dNTPs mix 10 mM, and 6.5 μL nuclease-free H_2_O, followed by an initial incubation at 65 °C for 5 min and subsequently chilling on ice for 1 min. Then, a mastermix solution containing 3 μL nuclease-free H_2_O, 4 μL from 5× RT buffer, 2 μL DTT 100 mM, and 1 μL of reverse-transcriptase enzyme 200 U/μL was added to the reaction mix, reaching a final volume of 20 μL. RT-PCR was performed at 42 °C for 15 min. The obtained cDNA was checked for its validity both in 1% agarose gel electrophoresis and in a Nanodrop Lite spectrophotometer (Thermo Fischer Scientific) at A260/A280 ratio, with an obtained OD > 1.7 in all samples. To further perform the gene expression analysis, a PCR for *REX1*, *OCT4*, *KLF4*, *NANOG*, *SOX9*, and *GAPDH* (Table 1) was performed in non-primed (*n* = 3) and IFN-γ-primed (*n* = 3) WJ-MSCs (in triplicate for each sample). The PCRs were performed in a gradient PCR (T Gradient Thermocycler, Biometra, Analytik Jena, Germany), involving the following steps: initial denaturation at 95 °C for 15 s, denaturation at 94 °C for 30 s, annealing at 59–61 °C (dependent on primers used) for 90s, and final extension at 72 °C for 30 s. The total number of cycles was 35, followed by electrophoresis on 2% agarose gel. For proper quantification of gene expression, real-time PCR was also performed using a Gentier 48E machine (Tianlong Science and Technology, Xi’an, China). The real-time PCR was performed using an RN014S kit (Enzyquest, Heraklion, Crete, Greece), following the manufacturer’s instructions, and it involved 40 cycles with the following steps: heat activation at 95 °C for 15 min, denaturation at 95 °C for 30 s, annealing at 59–61 °C for 30 s, and final extension 72 °C for 30 s. The quantification of gene expression analysis was performed using the formula ΔCt = Ct_g_ − Ct_ref_, and for the comparison between groups the formula ΔΔCT = ΔCt_non-primed_ − ΔCT_IFN-γ-primed_ was used. Fold change in gene expression was calculated using the formula 2^−ΔΔCt^.

### 4.14. HLA Typing

Initially, genomic DNA (gDNA) from MSCs (*n* = 20) was isolated using a DNeasy Blood and Tissue Kit (Qiagen, Hilden, Germany), following the manufacturer’s instructions. The concentration and purity of MSCs’ gDNA were estimated with a nanodrop spectrophotometer (260/280 nm). HLA typing for the HLA-A, -B, and -C and -DR, -DQ, and -DP were performed using an Omixon Holotype kit (Omixon, Budapest, Hungary), according to the manufacturer’s instructions. The analyses of the HLAs were performed using HLA Twin Software v3.1 (Omixon, Budapest, Hungary). Further analysis of HLA and neighbor-joining tree development for HLA class I and II were performed using Molecular Evolutionary Genetics Analysis (MEGA) 12 tool [72]. Nucleotide comparison of HLA class I and II alleles was performed using JalView version 2 [73].

### 4.15. Multivariate Data Analysis Using RStudio

The protein secretion profiles of non-primed and IFN-γ-primed WJ-MSCs were performed using the pheatmap package in RStudio v12.1. Clustering of non-primed and IFN-γ-primed WJ-MSCs based on their secretory profiles was performed using unsupervised machine learning algorithms (factominer, factoextra, umap). Hierarchical clustering of WJ-MSCs based on their HLA identification was performed using the factoextra and hclust packages.

### 4.16. Statistical Analysis

Statistical analysis was performed using GraphPad Prism v6 (GraphPad Software, San Diego, CA, USA). Statistically significant differences observed between study groups were indicated using the nonparametric Kruskal–Wallis test. Moreover, the validity of the results was further confirmed using the unpaired nonparametric Mann–Whitney U test. Statistically significant differences were considered when the *p*-value was less than 0.05. Indicated values were presented as mean ± standard deviation.

## 5. Conclusions

In conclusion, WJ-MSCs represent a valuable source of third-party stem cells, which can be applied in clinical trials for immune-related disorders to potentially reverse the pathological situation. WJ-MSCs can be non-invasively isolated and easily expanded in great numbers under standard in vitro culturing conditions. Moreover, stimulated WJ-MSCs by inflammatory signaling cues seem to retain their stemness and multipotency state for at least 96 h. In the context of a precise medicinal approach, high-resolution HLA typing of MSCs to select the most histocompatible cells for a specific recipient should be performed to avoid any possibility of patient sensitization. Therefore, the establishment of an MSC biobank, where well-defined MSC lines will be stored to be readily available for immediate use, presents great interest. This, in turn, may favor the broader application of MSCs in severe conditions such as in corticosteroid-resistant GVHD and other immune-related disorders, making them ideal ATMPs for clinical translation and utility.

## Figures and Tables

**Figure 1 ijms-26-09436-f001:**
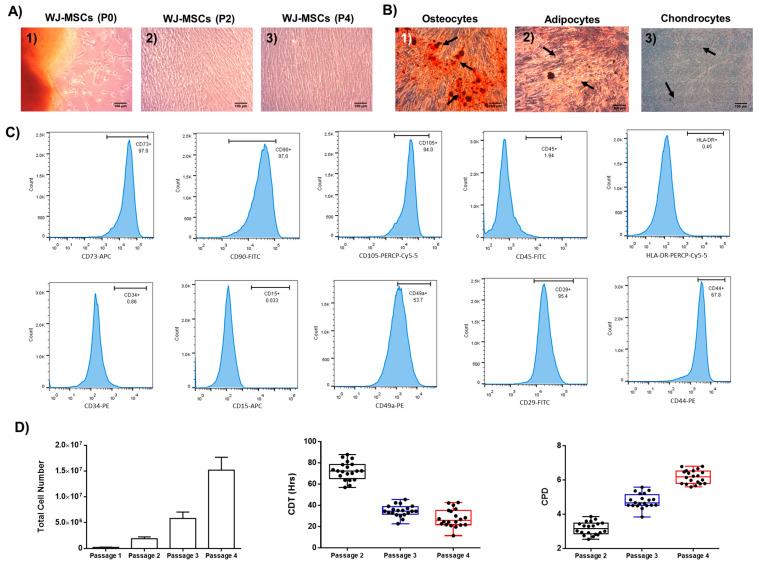
Evaluation of WJ-MSC characteristics based on the minimum criteria outlined by ISCT. WJ-MSCs presented fibroblastic-like morphology in passages P0 until reached P3 (**A**). Differentiation of WJ-MSCs to “osteocytes”, “adipocytes”, and “chondrocytes”, confirmed by alizarin red O, oil red O, and toluidine blue stains, respectively (**B**). Black arrows indicated the presence of calcium deposits, oil drops and glycosaminoglycans in “osteocytes”, “adipocytes” and “chondrocytes”, respectively. Immunophenotypic analysis of WJ-MSCs for CD73, CD90, CD105, CD45, HLA-DR, CD34, CD15, CD49a, CD29, and CD44 (**C**). Determination of WJ-MSC characteristics, including the total cell number, CDT, and CPD until it reached P4 (**D**). All images obtained with original magnification 10× and presented with scale bars 100 μm.

**Figure 2 ijms-26-09436-f002:**
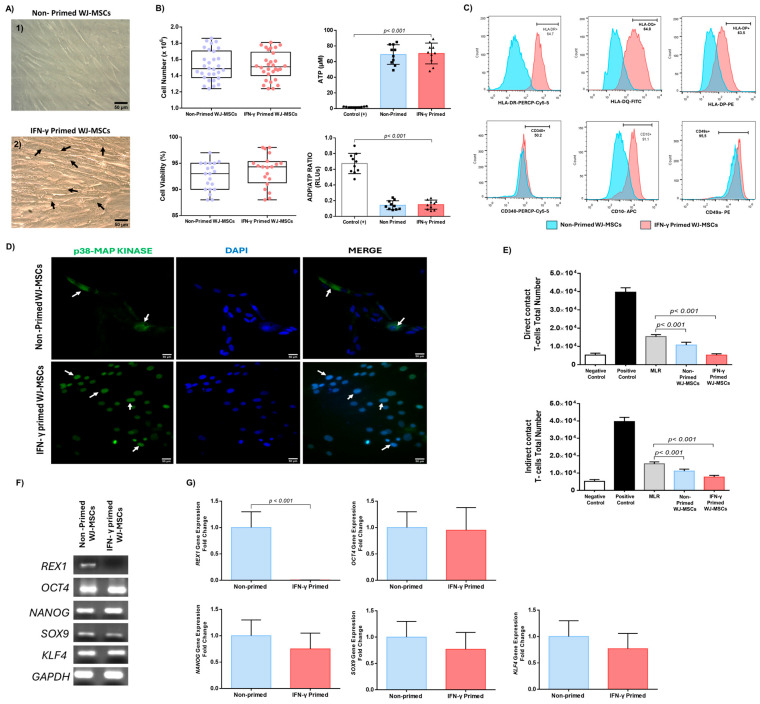
Impact of IFN-γ on MSCs’ phenotypic characteristics. Microscopic evaluation of non-primed WJ-MSCs and IFN-γ-primed WJ-MSCs (**A**). Black arrows indicate the presence of intracellular vesicles in IFN-γ-primed MSCs. Microscopic images were obtained with original magnification 20×, and are presented with scale bars of 50 μm. Total cell number, viability, total ATP, and ADP/ATP ratio measured in non-primed and IFN-γ-primed WJ-MSCs (**B**). No statistically significant differences were observed regarding the total cell number (*p* = 0.969), viability (*p* = 0.232), total ATP (*p* = 0.832), and ADP/ATP ratio (*p* = 0.999) between the non-primed and IFN-γ-primed MSCs. The only statistically significant differences were observed regarding the total ATP (*p* < 0.01) and ADP/ATP ratio (*p* < 0.01) between the positive control group and the rest of the study groups. Surface marker expression of HLA-DR, -DQ, -DP CD10, CD340, and CD49a of non-primed and IFN-γ-primed WJ-MSCs (**C**). Indirect immunofluorescence against p38 MAP kinase counterstained with DAPI in non-primed and IFN-γ-primed WJ-MSCs (**D**). White arrows indicated the presence of p38 MAP kinase P38 MAP kinase stain (green) was co-localized with DAPI stain (blue) in cell nuclei of IFN-γ-primed WJ-MSCs. Microscopic images were obtained with original magnification of 20× and 40×, and are presented with scale bars of 50 and 10 μm, respectively. MLR was conducted either in direct or indirect contact (**E**). Statistically significant reductions in CB-T cell numbers were observed when co-cultured with WJ-MSCs (both IFN-γ-primed and non-primed) in MLR experiments (*p* < 0.001). Gene expression analysis, regarding the *REX1*, *OCT4*, *NANOG*, *SOX9*, *KLF4,* and *GAPDH* in non-primed and IFN-γ-primed WJ-MSCS presented in gel electrophoresis (**F**). Fold change (2^−ΔΔCt^) of *REX1*, *OCT4*, *NANOG*, *SOX9*, *KLF4*, and *GAPDH* expression levels were assessed in non-primed and IFN-γ-primed WJ-MSCs (**G**). The gene *GAPDH* was utilized as a housekeeping control for the normalization of gene expression levels. A statistically significant difference in *REX1* expression (*p* < 0.001) was found between non-primed and IFN-γ-primed WJ-MSCs.

**Figure 3 ijms-26-09436-f003:**
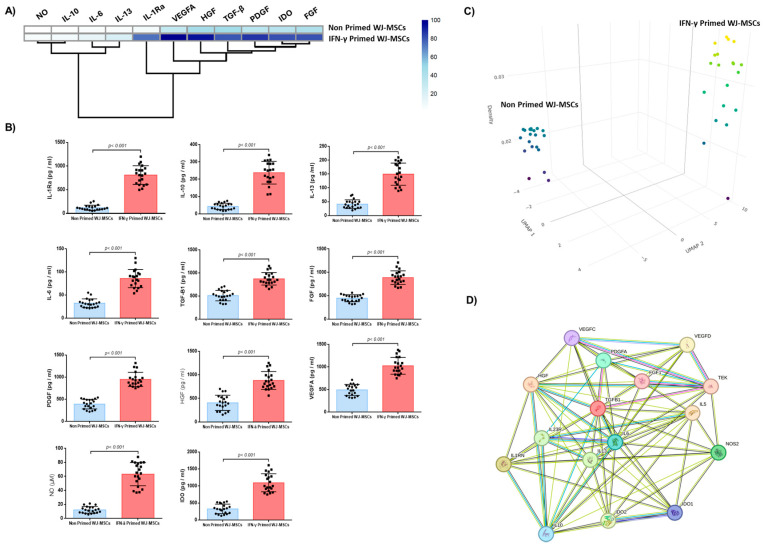
Biomolecules quantification of IFN-γ-primed and non-primed WJ-MSCs. Heatmap of secreted anti-inflammatory cytokines and growth factors in non-primed and IFN-γ-primed WJ-MSCs (**A**). Color intensity indicates the relative expression levels. Quantification of released cytokines, including IL-1Ra, IL-10, IL-13, and IL-6, growth factors including TGF-β1, FGF, PDGF, HGF, and VEGFA, and biomolecules including NO and IDO by non-primed and IFN-γ-primed WJ-MSCs (**B**). Statistically significant differences were observed in all quantified biomolecules between non-primed and IFN-γ-primed WJ-MSCs (*p* < 0.001). Three-dimensional UMAP plot depicts the levels of biomolecule secretion by IFN-γ-primed WJ-MSCs and non-primed WJ-MSCs (**C**). String protein–protein interaction network illustrating the synergistic association of the secreted biomolecules (**D**).

**Figure 4 ijms-26-09436-f004:**
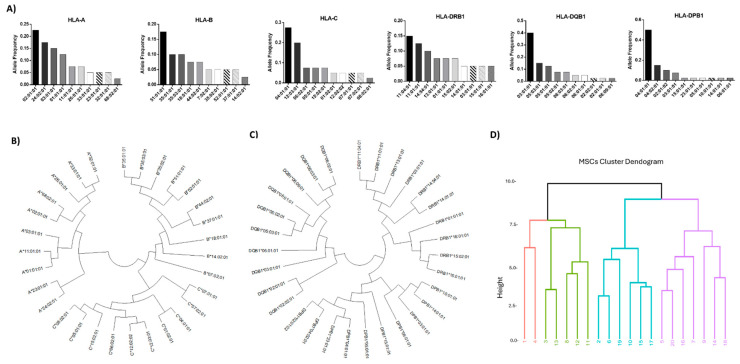
Determination of HLA class I and II allele frequencies of WJ-MSCs. HLA class I (HLA-A, -B, and -C) and class II (HLA-DPB1, -DQB1 and -DRB1) frequencies in WJ-MSCs (**A**). Neighbor-joining tree of the most frequent HLA class I (**B**) and class II (**C**) alleles. Hierarchical clustering of WJ-MSCs based on HLA class I and II frequencies and the biomolecule secretory profile (**D**).

**Figure 5 ijms-26-09436-f005:**
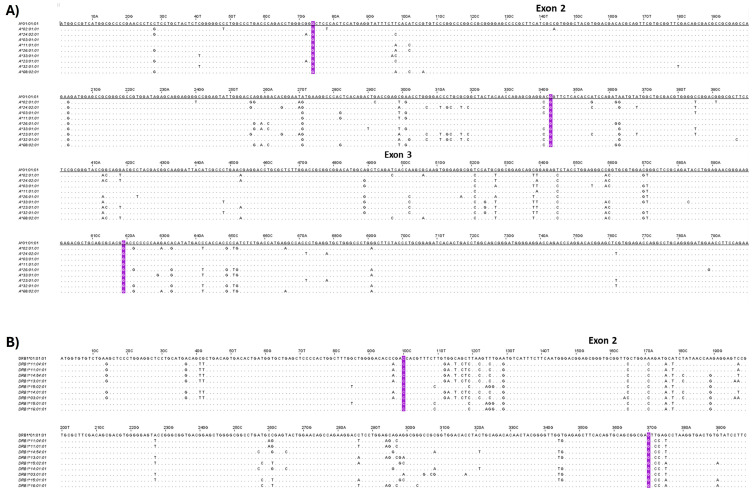
Representative images of HLA class I (HLA-A alleles) and class II (HLA-DRB1 alleles) nucleotide comparisons regarding exons 2 and 3. Multiple nucleotide changes were observed in exons 2 and 3 within the different alleles of HLA-A (**A**) and in exon 2 within the different alleles of HLA-DRB1 (**B**).

**Table 1 ijms-26-09436-t001:** Primer pairs used for gene expression analysis.

Gene	Forward Sequence	Location	Reverse Sequence	Location	Amplicon Size
*REX1*	CCCTGGAATACGTCCCCAAG	718–737	CATCCTGTGAGGACTGGACC	1227–1208	510
*OCT4*	GTGTTCAGCCAAAAGACCATCT	532–553	GGCCTGCATGAGGGTTTCT	687–669	156
*NANOG*	TTTGTGGGCCTGAAGAAAACT	83–103	AGGGCTGTCCTGAATAAGCAG	198–178	116
*KLF4*	CGGACATCAACGACGTGAG	563–581	GACGCCTTCAGCACGAACT	701–683	139
*SOX9*	AGCGAACGCACATCAAGAC	1178–1196	CTGTAGGCGATCTGTTGGGG	1262–1243	85
*GAPDH*	GGAGCGAGATCCCTCCAAAAT	108–128	GGCTGTTGTCATACTTCTCATGG	304–282	197

## Data Availability

The original contributions presented in this study are included in the article/Appendix A. Further inquiries can be directed to the corresponding author(s).

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
