# Peer review of "The Impact of IFN-γ Licensing on Mesenchymal Stromal Cells’ Mediated Immunoregulation and HLA Class II Expression: Emerging Evidence from In Vitro Results"

_ijms, 2025, doi:10.3390/ijms26199436_

Round 1
Reviewer 1 Report
Comments and Suggestions for Authors
The manuscript is overall clear, although it would benefit from a revision by a native EN speaker. The material adds limited novelty to the state of the art, although it can be considered relevant for it validates assumptions that could be applied to MSC-based ATMP use. The data are presented in a well-structured manner in terms of sequence. The cited references do not consist of recent publications at large but are considered relevant in number and with an appropriate number of self-citations. The manuscript is scientifically sound and is the experimental design appropriate to test the hypothesis - although presenting itself poor in novelty with over 90% material being a repetition and validation of many previous studies. The manuscript’s results are convincingly reproducible based on the details given in the methods section, which provided approtiate detail. The figures/tables/images/schemes appropriate are globally appropriate. However, references to tables that are under supplementary data should refer that when cited - the reader will look for them in the main text if this is not done. However in many cases graph and figure legends are too small and difficult to read. This should revised for publication. Nevertheless, figures properly show the data and are easy to interpret and understand for someone skillful in the area. Ethics aspects and data availability aspects are well coverd. Data interpreted appropriately and consistently throughout the manuscript. The conclusions are consistent with the evidence and arguments presented. In general terms, the paper is relatively poor in innovation as many previous studies have demonstrated that MSCs (namely WJMSC) can modify their characteristics, notably exhibit increased expression of HLA class II molecules, when activated by inflammatory stimuli, in particular by interferon-gamma (IFN-γ). Nevertheless, on the clustering analysis which indicates that MSCs can be classified according to their secretory profiles and HLA allele frequencies into groups reflecting high, moderate, and low responses is interesting. As a future perspective, this could indeed lead to the establishment of a cell biobank with well-defined MSC lines, to be readily accessible upon demand in a more personalized manner. The introduction of NGS and the demonstration of its applicability is plus.
Author Response
Dear Reviewer 1,
Below you can find our responses towards your comments.
|
General Comment |
|
The manuscript is overall clear, although it would benefit from a revision by a native EN speaker. The material adds limited novelty to the state of the art, although it can be considered relevant for it validates assumptions that could be applied to MSC-based ATMP use. The data are presented in a well-structured manner in terms of sequence. The cited references do not consist of recent publications at large but are considered relevant in number and with an appropriate number of self-citations. The manuscript is scientifically sound and is the experimental design appropriate to test the hypothesis - although presenting itself poor in novelty with over 90% material being a repetition and validation of many previous studies. The manuscript’s results are convincingly reproducible based on the details given in the methods section, which provided approtiate detail. The figures/tables/images/schemes appropriate are globally appropriate. However, references to tables that are under supplementary data should refer that when cited - the reader will look for them in the main text if this is not done. However in many cases graph and figure legends are too small and difficult to read. This should revised for publication. Nevertheless, figures properly show the data and are easy to interpret and understand for someone skillful in the area. Ethics aspects and data availability aspects are well coverd. Data interpreted appropriately and consistently throughout the manuscript. The conclusions are consistent with the evidence and arguments presented. In general terms, the paper is relatively poor in innovation as many previous studies have demonstrated that MSCs (namely WJMSC) can modify their characteristics, notably exhibit increased expression of HLA class II molecules, when activated by inflammatory stimuli, in particular by interferon-gamma (IFN-γ). Nevertheless, on the clustering analysis which indicates that MSCs can be classified according to their secretory profiles and HLA allele frequencies into groups reflecting high, moderate, and low responses is interesting. As a future perspective, this could indeed lead to the establishment of a cell biobank with well-defined MSC lines, to be readily accessible upon demand in a more personalized manner. The introduction of NGS and the demonstration of its applicability is plus. |
|
Author’s Response |
|
Initially, we would like to gratefully thank the reviewer for the valuables comments provided herein. We agree with the fact that also previously it was mentioned by Katerina Le Blanc the intracellular expression of HLA class II in MSCs. However, there is limited literature worldwide to present clear evidence regarding the upregulation of HLA class II in MSCs upon priming by inflammatory stimulis e.g. IFN-γ. In the submitted study this was clearly presented and can be easily reproduced by the researchers in the field. However, the advantage of the work is relied how the different HLAs based on their frequencies in human population can actually play important role in the selection of MSCs. Nowadays, MSCs are considered ATMPs (EU Regulation No 1394/2007). In addition, when MSCs will be administrated in patients suffering from immune related disorders, to avoid any sensitization adverse events and to exert the greatest immunomodulatory potential, the HLA histocompatibility between donor and recipient should be strongly considered. In literature, there is evidence that kidney transplanted patients sensitized and formed DSAs against the foreign HLAs presented by the all-MSCs. Therefore for the development and administration of a safe and well tolerable ATMP therapy utilizing the MSCs, the high resolution HLA typing with the NGS technology and their clustering based on their secretory profile and HLAs is of great importance. Hence the physicians can select easier the MSCs based on the genetic background of their patients. Therefore the innovation of this work relies on the greatest understanding of MSCs immunobiology, in order to manipulated them properly and to form a biobank with well defined cell lines. This strategic plan will target the unmet need of production of ready to use third party MSCs to combat severe human disorders including the autoimmune diseases, GvHD, cancer., etc. Regarding the language polishing, typos and grammar or phrase errors the whole manuscript has been extensively revised by an English native speaker of our team. |
We think that with the performed revisions, the quality of the manuscript has been increased and can be further processed to the next step of the publication process.
We remain at your disposal if anything else is required by our side.
Yours sincerely,
Panagiotis Mallis, MSc, PhD
GMP Lab Technical Supervisor
Hellenic Cord Blood Bank,
Biomedical Research Foundation Academy of Athens,
4 Soranou Efessiou Street,
11527 Athens, Greece
tel:+302106597340, mob:+306971616467
e-mail: pmallis@bioacademy.gr

Reviewer 2 Report
Comments and Suggestions for Authors
In this paper, the authors study the effects of INF-γ priming on Wharton’s jelly-derived mesenchymal stromal cells’ (WJ-MSCs) phenotypic and immunoregulatory properties. Since IFN-γ priming induces MHC II expression in WJ-MSCs, the authors argue, justly, that this could render them immunogenic and targeted for clearance by the host immune system. For that reason, the authors evaluated the HLA polymorphism, as well as the secretory profile of WJ-MSCs from 20 donors. This is an important issue being raised, which deserves more attention in light of the growing number of actual and potential applications of MSCs as advanced therapeutic medicinal products. Hence, the study is impactful and offers some novel insights into the field of applied MSC research. On the other hand, the manuscript has some deficiencies that need to be addressed. Some parts, especially in the introduction, are confusing and misleading. A thorough revision is warranted to improve the language and clarity of writing. Some formatting issues occur, as well. I have left some comments and questions in the enclosed PDF file that should be addressed before proceeding with the manuscript being accepted for publication. Some of the key issues are the following:
- There are a few erroneous and misleading statements in the introduction, like that AF-MSCs are embryonic stem cells, that T cells are cells of the innate immunity, or that calcineurin inhibitors are steroids…
- In the figure legends, some important parts are missing. In Figure 1, part D is left unexplained, while in part C (which is noted as B), CD10 and CD340, which are not in the figure, are named instead of CD49a and CD34, which are. In Figure 2 legend, it is also confusing which part of the legend refers to which part of the figure. In Figure 3 legend, nothing except part B is explained.
- In Figure 2G, showing qPCR results, I would suggest using 2^-ΔΔCt (fold change, as in supplementary Figure S3) instead of ΔCt, since it is more straightforward when the IFN-γ-primed column for Rex1 is lower than the non-primed, as the actual expression of the gene is lower.
- The results are well discussed, and the implications for clinical practice are explained, but the authors should also mention the study’s limitations and address the prospects for future research.
- In Table 1 (Materials and Methods), data for SOX9 primers are missing.
- Reference 41 is the same as 40. Remove it and correct the numbering in the text accordingly.
- Please refer to the uploaded PDF file for other questions, comments, and suggestions. Pay attention to the insertions and deletions, as those are less notable than the highlights in the text.

The language quality and the clarity of writing should be significantly improved. I would suggest consulting a native English speaker or using a professional proofreading service.
Author Response
Dear Reviewer 2,
We would like to thank the reviewer for the comments performed to our manuscript. We have revised the manuscript based on the reviewer’s comments and below you can find our step by step responses. All changes performed in the revised manuscript have been indicated with the track changes tool of the Microsoft word tool.
|
General Comments |
|
In this paper, the authors study the effects of INF-γ priming on Wharton’s jelly-derived mesenchymal stromal cells’ (WJ-MSCs) phenotypic and immunoregulatory properties. Since IFN-γ priming induces MHC II expression in WJ-MSCs, the authors argue, justly, that this could render them immunogenic and targeted for clearance by the host immune system. For that reason, the authors evaluated the HLA polymorphism, as well as the secretory profile of WJ-MSCs from 20 donors. This is an important issue being raised, which deserves more attention in light of the growing number of actual and potential applications of MSCs as advanced therapeutic medicinal products. Hence, the study is impactful and offers some novel insights into the field of applied MSC research. On the other hand, the manuscript has some deficiencies that need to be addressed. Some parts, especially in the introduction, are confusing and misleading. A thorough revision is warranted to improve the language and clarity of writing. Some formatting issues occur, as well. I have left some comments and questions in the enclosed PDF file that should be addressed before proceeding with the manuscript being accepted for publication. Some of the key issues are the following: |
|
Reviewer’s Comments |
|
1. There are a few erroneous and misleading statements in the introduction, like that AF-MSCs are embryonic stem cells, that T cells are cells of the innate immunity, or that calcineurin inhibitors are steroids… |
|
Author’s Response |
|
The reviewer is right, there are some phrase errors in the entire manuscript. The whole manuscript has been extensively revised for such errors. Regarding the errors that the reviewer indicated, the following sentences regards the changes that have been performed in the manuscript as follows: In the introduction section-AF-MSCs are embryonic stem cells: Based on their origin, MSCs can be distinguished into, fetal stem cells, e.g. AF-MSCs and WJ-MSCs and adult stem cells, e.g. BM-MSCs, AT-MSCs, and SVF-MSCs.” In the introduction section-T cells are cells of the innate immunity: “Once antigenic epitope presentation occurs by the professional antigen-presenting cells (APCs), then recognition by the cells of adaptive immunity, e.g. CD4+ or CD8+ T cells, is followed, resulting in the secretion of high amounts of pro-inflammatory cytokines such as IFN-γ, TNF-α and IL-1β, chemokines originating from the damaged cells and tissues, such as CCL2, CXCL10, CCL19 and pro-inflammatory cytokines, e.g. TNF-α, IL-1β andIL-3 and [14]” In the introduction section- calcineurin inhibitors are steroids: “To reverse the adverse events of aGvHD, patients mostly receive steroids such as prednisone or methylprednisolone , however, a number of patients are less responsive to this therapeutic strategy, resulting in the development of steroid-resistant (SR) condition [21].” |
|
Reviewer’s Comments |
|
2. In the figure legends, some important parts are missing. In Figure 1, part D is left unexplained, while in part C (which is noted as B), CD10 and CD340, which are not in the figure, are named instead of CD49a and CD34, which are. In Figure 2 legend, it is also confusing which part of the legend refers to which part of the figure. In Figure 3 legend, nothing except part B is explained. |
|
Author’s Response |
|
We thank the reviewer for this comment. We have revised all figures legends in the whole manuscript. Below you will find the performed revisions. Figure 1. Evaluation of WJ-MSCs characteristics based on the minimum criteria outlined by ISCT. WJ-MSCs presented fibroblastic-like morphology, which was retained from P0 (A1), to P1(A2) and P3(A3). Immunophenotypic analysis of WJ-MSCs for CD73, CD90, CD105, CD45, HLA-DR, CD34, CD15, CD49a, CD29 and CD44.CD29, CD10, CD340, CD44 and CD15 (B). Successful differentiation of WJ-MSCs to “osteocytes”, “adipocytes” and “chondrocytes”, as it was confirmed by Alizarin Red O, Oil-Red O and Toluidine blue stains, respectively. Determination of WJ-MSCs characteristics including the total cell number, CDT and CPD until reached P4, (D). All images obtained with original magnification 10x and presented with scale bars 100 μm. Figure 2. Impact of IFN-γ on MSCs' phenotypic characteristics. Microscopic evaluation of non-primed WJ-MSCs (A1) and IFN-γ primed WJ-MSCs (A2). Black arrows in image A2 indicated the presence of intracellular vesicles in IFN-γ primed MSCs. Microscopic images were obtainedpresented with original magnification 20x, and pressented with scale bars 50 μm. Total cell number, viability, total ATP and ADP/ATP ratio measured in non-primed and IFN-γ primed WJ-MSCs (B). No statistically significant differences were observed regarding the total cell number (p = 0.969), viability (p = 0.232), total ATP (p = 0.832) and ADP/ATP ratio (p = 0.999) between the non-primed and IFN-γ primed MSCs. The only statistically significant difference was observed regarding the total ATP (p < 0.01) and ADP/ATP ratio (p < 0.01) between the positive control group and the rest of theother study groups. Surface marker expression of HLA-DR, CD10, CD340, and CD49a of non-primed and IFN-γ primed WJ-MSCs (C). Indirect immunofluorescence against p38 MAP kinase counterstained with DAPI, in non-primed and IFN-γ primed WJ-MSCs (D). In IFN-γ primed WJ-MSCs, Pp38 MAP kinase stain (green) was co-localizedobserved in combination with DAPI stain (blue) in cell nuclei of IFN-γ primed WJ-MSCs . Microscopic images were obtainedpresented with original magnification of 20x and 40x, and presented with scale bars of 50 and 10 μm, respectively. MLR was conductedperformed either in direct or indirect contact (E)with ΙFN-γ primed and non-primed WJ-MSCs. A statistically significant reduction in CB-T cell numbers was observed when co-cultured with WJ-MSCs (both IFN-γ primed and non-primed) in MLR experiments (p < 0.001). Statistically significant differences were observed between the IFN-γ primed WJ-MSCs and non-primed WJ-MSCs either in direct (p < 0.01) or in direct contact (p < 0.01) MLR. Gene expression analysis, regarding the REX1, OCT4, NANOG, SOX9, KLF4 and GAPDH in non-primed and IFN-γ primed WJ-MSCS presented in gel electrophoresis (F). Fold change (2^-ΔΔCt) of REX1, OCT4, NANOG, SOX9, KLF4, and GAPDH expression levels were assessed in non-primed and IFN-γ primed WJ-MSCs (G). The gene GAPDH was utilised as a housekeeping control for the normalization of gene expression levels. Performance of real-time PCR for the evaluation of gene expression, regarding the REX1, OCT4, NANOG, SOX9, KLF4 and GAPDH. GAPDH served as a house-keeping gene. A statistically significant difference in REX1 expression (p < 0.001) was found between non-primed and IFN-γ primed WJ-MSCs. Statistically significant difference was observed between non-primed and IFN-γ primed WJ-MSCs regarding the REX1 (p < 0.001). Figure 3. Biomolecules quantification of IFN-γ primed and non-primed WJ-MSCs. Heatmap of secreted anti-inflammatory cytokines and growth factors in non primed and IFN-γ primed WJ-MSCs (A). Color intensity indicated the relative expression levels. Quantification of releasedof anti-inflammatory cytokines, including the IL-1Ra (A),IL-10, IL-13, IL-6, growth factors including IL-6 (B), IL-10 (C) and IL-13 (D), exerted by non-primed and IFN-γ primed WJ-MSCs. Quantification of growth factors involved in immunoregulation, including the TGF-β1 (Ε), FGF (F), PDGF (D), VEGFA (H), HGF, VEGFA, a (I), and biomolecules including NO and IDO produced by non-primed and IFN-γ primed WJ-MSCs (B). Quantification of other immunomodulatory biomolecules, such as IDO and NO, exerted from non-primed and IFN-γ primed WJ-MSCs. Statistically significant difference was observed in all quantified biomolecules between non-primed and IFN-γ primed WJ-MSCs (p < 0.001). Three dimensional UMAP plot depicted the levels of biomolecules secretion by IFN-γ primed WJ-MSCs and non primed WJ-MSCs (C). String protein-protein interaction network illustrating the synergistic association of the secreted biomolecules (D). Figure 4. Determination of HLA class I and II allele frequencies of WJ-MSCs. HLA class I (HLA-A, -B, and – C) and class II (HLA-DPB1, -DQB1 and -DRB1) frequencies in WJ-MSCs (A). Neighbor joining tree of the most frequent HLA class I, including the HLA-A, HLA-B and HLA-C (B) and class II, including the HLA-DRB1, HLA-DQB1 and HLA-DPB1 (C) alleles. Hierarchical clustering of WJ-MSCs based on the HLA class I and II frequencies and biomolecule secretory profile (D). Figure 5. Representative images of HLA class I (HLA-A alleles) and class II (HLA-DRB1 alleles) nucleotide comparison regarding exons 2 and 3. Multiple nucleotide changes were observed in exons 2 and 3 within the different alleles of HLA-A (A) and in exon 2 within the different alleles of HLA-DRB1 (B). |
|
Reviewer’s Comments |
|
3. In Figure 2G, showing qPCR results, I would suggest using 2^-ΔΔCt (fold change, as in supplementary Figure S3) instead of ΔCt, since it is more straightforward when the IFN-γ-primed column for Rex1 is lower than the non-primed, as the actual expression of the gene is lower. |
|
Author’s Response |
|
Again, we would like to thank the reviewer for this valuable comment. We have entered the fold change (2^-ΔΔCt) of all gene expression levels in figure 2. The revised figure 2 has been inserted to the manuscript. In addition, the initial ΔCT diagrams have been inserted in the supplementary file. |
|
|
|
4. The results are well discussed, and the implications for clinical practice are explained, but the authors should also mention the study’s limitations and address the prospects for future research. |
|
Author’s Response |
|
To address the limitations and the future perspectives of the study the following paragraph was added in the discussion section. “In this study, it was clearly shown that IFN-γ primed MSCs secrete key immunomodulatory biomolecules, potentially to contribute to the toleration of the overstimulated immune responses, however a significant upregulation of HLA class II was also noted. Since HLA class II may trigger the development of DSAs in the host, this finding may be further correlated with impaired immune toleration exerted by MSCs. To further elucidate the possibility of functional consequences of HLA class II expressed in MSCs when administrated in patients suffering from immune-related disorders, in vivio studies with specific animal models are required. This fact represents the main limitation of the the present work, which will be explored in the near future, taking into account the results presented herein. Specifically, autoimmune humanized animal models should be employed to assess the functional relevance of HLA class II, in DSAs formation. Overall, the obtained results, provide new insights to understand better the immunobiology of MSCs, which can be potentially utilized as alternative therapeutic strategy to combat the human immune-related disorders.” |
|
Reviewer’s Comments |
|
5. In Table 1 (Materials and Methods), data for SOX9 primers are missing |
|
Author’s Response |
|
The reviewer is right. The primer pair for SOX9 has been included in table 1 of the materials and method section. |
|
Reviewer’s Comments |
|
6. Reference 41 is the same as 40. Remove it and correct the numbering in the text accordingly. |
|
Author’s Response |
|
The reviewer was right. The double inserted ref “Yang, Z.X.; Mao, G.X.; Zhang, J.; Wen, X.L.; Jia, B.B.; Bao, Y.Z.; Lv, X.L.; Wang, Y.Z.; Wang, G.F. IFN-γ Induces Senescence-like Characteristics in Mouse Bone Marrow Mesenchymal Stem Cells. Adv Clin Exp Med 2017, 26, 201–206, doi:10.17219/ACEM/61431” listed as 40 and 41 corrected using the Mendeley. All references checked again and validated for their proper place to be represented only once in the list. |
|
Reviewer’s Comments |
|
7. Please refer to the uploaded PDF file for other questions, comments, and suggestions. Pay attention to the insertions and deletions, as those are less notable than the highlights in the text. |
|
Author’s Response |
|
We gratefully thank the reviewer for the comprehensive peer-review performed to our submitted work. We have checked again the whole manuscript for any typos, grammar and phrase errors and revised where it was necessary. |
|
Reviewer’s Comments |
|
8. Beyond the above revisions, the whole manuscript has been checked for any grammar and phrase errors by a native English speaker and revised accordingly as has been suggested by the reviewers. |
|
Author’s Response |
|
The whole manuscript has been properly rephrased and the language has been polished by a native English speaker of our team. Typos, grammar or phrase errors have been revised accordingly throughout the whole manuscript. |
We think that with the performed revisions, the quality of the manuscript has been increased and can be further processed to the next step of the publication process.
We remain at your disposal if anything else is required by our side.
Yours sincerely,
Panagiotis Mallis, MSc, PhD
GMP Lab Technical Supervisor
Hellenic Cord Blood Bank,
Biomedical Research Foundation Academy of Athens,
4 Soranou Efessiou Street,
11527 Athens, Greece
tel:+302106597340, mob:+306971616467
e-mail: pmallis@bioacademy.gr

Reviewer 3 Report
Comments and Suggestions for Authors
The manuscript considers the reply to the following question: "Could the immunoregulatory properties of MSCs be compromised by the expression of HLA class II upon IFN-γ licensing?"
To this aim, the authors have isolated Wharton’s Jelly (WJ)-MSCs from the human umbilical cords. The cells, after a brief period of culture, were exposed to IFN (100 ng/μl IFN-γ) or not. After 96 h, the IFN-γ-primed WJ-MSCs were used for the subsequent experiments.
Proliferation was assessed by ATP content evaluation, and WJ-MSCs cell death by ADP/ATP ratio assay. Biomolecules, including the anti-inflammatory cytokines and growth factors produced from IFN-γ-primed WJ-MSCs, were quantified using commercial ELISA assay kits. Furthermore, MLR of cord blood FACS-isolated T cells was performed either in direct or indirect contact with WJ-MSC. The stimulation index was determined, and the positive control was represented by the same combination stimulated with PHA. The HLA-typing has also been performed.
The IFN-γ-primed WJ-MSCs neoexpressed HLA-II and CD10. The WJ-MSC treated with IFN released more IL-1Ra, IL-6, IL-10, and IL-13 among the cytokines than untreated cells. TGF-β1, FGF, PDGF, VEGFA, HGF, NO, and IDO of the IFN-γ-primed WJ-MSCs present in culture supernatants were more than those present in untreated WJ-MSCs. Importantly, these effects were not evident in 25% of the 20 samples analyzed. This indicates a certain heterogeneity among the different WJ-MSC preparations.
The authors also stated that the functional protein association network showed that IL-6 and TGF-β1 were the main orchestrators for anti-inflammatory responses and actively interacted and possibly regulated the production of the immunoregulatory biomolecules secreted by the IFN-γ-WJ-MSCs.
The authors conclude that WJ-MSc can be isolated and expanded easily to reach high numbers to be used in a clinical setting. These cells maintain stemness properties, and it is relevant to define the HLA-I and HLA-II expressed by these cells to select the most histocompatible cells for a specific donor. This should be performed to avoid any possibility of patient sensitization and rapid clearance of cells.
The major strength of this work is the description of the response of 20 different samples of WJ-MSC. The overall message is similar to several papers published in the literature using this kind of MSC or MSC from other tissues (bone marrow, adipose tissue, skin, and others).
The analysis of HLA-typing of WJ-MSc is of interest, and in particular, the upregulation by IFN of HLA-II. The biological significance and function of this upregulation are not tested. Indeed, the expression of HLA-II antigens can be relevant for antigen presentation, but this point is not addressed in this work. It is unclear whether the observed upregulation can influence the immunoregulatory properties of WJ-MSC.
The upregulation of HLA-II is tested with a single anti-HLA class II antibody; at least another one to confirm this event should be considered. It has not tested the upregulation of relevant molecules such as ICAMs and PDL1 (also CD80 and CD86 can be of interest to further define the relevance of the cells analyzed). These molecules are essential for the immune response.
The authors consider in different portions of this work IL1b and IL6 as immunoregulatory molecules, in some others as immunoinflammatory molecules (see introduction section).
To my knowledge (and looking at the original definition of IL1b and IL6), these cytokines are essential for the inflammatory response. Please be consistent throughout the manuscript. IL6 can have different effects on different cell types. But in this case, please clarify better.
Specific concerns
1- The dot plot of Figure 1 panel C should contain the same number of events. This is not the case, evidently.
2- The finding/observation that IFN-γ primed-WJ-MSCs show some increase in intracellular vesicles should be demonstrated by conventional and/or advanced intracellular microscopy examination. The data shown does not allow to state what the authors stated.
3- The upregulation of HLA-II should be shown in all the IFN-γ primed-WJ-MSCs and corresponding controls. The example presented in panel C is informative, but the strength of this paper is in the number of samples analyzed, and this should be stressed.
4- The cell death of IFN-γ primed-WJ-MSCs should be shown with an additional methods that indeed identify dying/apoptotic cells.
5- MLR is performed (apparently) by mixing T cells separated from two cord blood donors. The T cells have been separated by FACS as indicated in the M&M section. The method of separation may trigger the CD3/TCR complex. A negative selection system of separation should be used to avoid the CD3-mediated signalling. The absence of monocytes could impair/alter the MLR response. Further, the presence of antigen-presenting cells is essential for an appropriate MLR. Please clarify. The analysis after 72 hours of MLR is quite short. Usually, the MLR can give a good response in 5-7 days of stimulation. Please explain.
6- It is unusual that the PHA response is almost superimposable to MLR (panel E, figure 2) tested at 72 hours. Usually, PHA can give a 10-20 fold stronger response compared to MLR. Indeed, the cells that can respond to MLR are usually 1 out of 10,0, while the large majority of T cells can respond to PHA (almost 100 out of 100).
7- It would be of interest to define whether the frequency of the different HLA class I and II alleles is in line with the frequency of these alleles in the whole population (not only in the WJ-MSCs).
8- The functional relevance of the upregulation of HLA-II as well as HLA-I on IFN-γ primed-WJ-MSCs should be tested.
Author Response
Dear Reviewer 3,
We would like to thank the reviewer for the comments performed to our manuscript. We have revised the manuscript based on the reviewer’s comments and below you can find our step by step responses. All changes performed in the revised manuscript have been indicated with the track changes tool of the Microsoft word.
|
1. General Comments |
|
The manuscript considers the reply to the following question: "Could the immunoregulatory properties of MSCs be compromised by the expression of HLA class II upon IFN-γ licensing?" To this aim, the authors have isolated Wharton’s Jelly (WJ)-MSCs from the human umbilical cords. The cells, after a brief period of culture, were exposed to IFN (100 ng/μl IFN-γ) or not. After 96 h, the IFN-γ-primed WJ-MSCs were used for the subsequent experiments. Proliferation was assessed by ATP content evaluation, and WJ-MSCs cell death by ADP/ATP ratio assay. Biomolecules, including the anti-inflammatory cytokines and growth factors produced from IFN-γ-primed WJ-MSCs, were quantified using commercial ELISA assay kits. Furthermore, MLR of cord blood FACS-isolated T cells was performed either in direct or indirect contact with WJ-MSC. The stimulation index was determined, and the positive control was represented by the same combination stimulated with PHA. The HLA-typing has also been performed. The IFN-γ-primed WJ-MSCs neoexpressed HLA-II and CD10. The WJ-MSC treated with IFN released more IL-1Ra, IL-6, IL-10, and IL-13 among the cytokines than untreated cells. TGF-β1, FGF, PDGF, VEGFA, HGF, NO, and IDO of the IFN-γ-primed WJ-MSCs present in culture supernatants were more than those present in untreated WJ-MSCs. Importantly, these effects were not evident in 25% of the 20 samples analyzed. This indicates a certain heterogeneity among the different WJ-MSC preparations. The authors also stated that the functional protein association network showed that IL-6 and TGF-β1 were the main orchestrators for anti-inflammatory responses and actively interacted and possibly regulated the production of the immunoregulatory biomolecules secreted by the IFN-γ-WJ-MSCs. The authors conclude that WJ-MSc can be isolated and expanded easily to reach high numbers to be used in a clinical setting. These cells maintain stemness properties, and it is relevant to define the HLA-I and HLA-II expressed by these cells to select the most histocompatible cells for a specific donor. This should be performed to avoid any possibility of patient sensitization and rapid clearance of cells. The major strength of this work is the description of the response of 20 different samples of WJ-MSC. The overall message is similar to several papers published in the literature using this kind of MSC or MSC from other tissues (bone marrow, adipose tissue, skin, and others). |
|
1. Author’s Response |
|
Initially, we would like to thank the reviewer for the comprehensive and well performed to our submitted manuscript. We think that with the our responses listed below can revise the manuscript appropriately. |
|
2. Comments |
|
a. The analysis of HLA-typing of WJ-MSc is of interest, and in particular, the upregulation by IFN of HLA-II. The biological significance and function of this upregulation are not tested. Indeed, the expression of HLA-II antigens can be relevant for antigen presentation, but this point is not addressed in this work. It is unclear whether the observed upregulation can influence the immunoregulatory properties of WJ-MSC. b. The upregulation of HLA-II is tested with a single anti-HLA class II antibody; at least another one to confirm this event should be considered. It has not tested the upregulation of relevant molecules such as ICAMs and PDL1 (also CD80 and CD86 can be of interest to further define the relevance of the cells analyzed). These molecules are essential for the immune response. c. The authors consider in different portions of this work IL1b and IL6 as immunoregulatory molecules, in some others as immunoinflammatory molecules (see introduction section). To my knowledge (and looking at the original definition of IL1b and IL6), these cytokines are essential for the inflammatory response. Please be consistent throughout the manuscript. IL6 can have different effects on different cell types. But in this case, please clarify better. |
|
2. Author’s Response |
|
a. The reviewer is right, we have not tested how the expression of the HLA class II may impact the immunoregulatory properties of WJ-MSCs. Unlike previously published works, we have shown that IFN-γ primed WJ-MSCs are characterized by elevated expression of HLA class II. This means that if WJ-MSCs are required to be administered as alternative therapeutic strategy in immune related disorders (e.g. autoimmune disease, GvHD), the inflammatory microenvironment will eventually trigger the HLA class II expression in infused MSCs. In this way, if the HLA histocompatibility has not been taken into account in third-party (allogeneic) MSCs, there is a possibility that the host’s immune system recognizes them, initiating the immune response to result rapid clearance of MSCs before exerting their immunoregulatory properties. In this patients, the infused MSCs will be characterized by impaired immunoregulatory function, however, this will be very different to observed in patients. In the study of Bezstarosti et . al. Front Genet. 2024 Sep 27;15:1436194, provided evidence that kidney transplanted patients (n=3) developed DSAs against the allogeneic infused MSCs. It should be also accounted that kidney transplanted patients are characterized as severy conditioned, therefor the sensitization towards the foreign HLAs of the infused MSCs may be resulted due to this condition. However, to test it properly that allogeneic third party MSCs can elicin immune responses and finally sensitization, clinical trials should be performed to acquire safe results. Our study may help in understanding better the immunobiology of MSCs and the results may be beneficial to orchestrate a clinical study or to carafeully take in into account from the physisians when they decide to use third-party MSCs to their patients. One of our future experiments actually involves to test this hypothesis using animal models, including healthy controls and models with autoimmune encephalitis. Also, another experiment pointing out this is to use humanized animal models. The evaluation of the immune responses and MSCs clearance will be validated with flow cytometry, histological analysis and mass cytometry. However, we think that our initial results are very important to be published, before performing the assessment in the animal models. In addition, acquiring and to perform this assessment in animal models will eventually take beyond one year of experiments to conclude to safe results. However, these experiments will be included as part of the experimental study of our next publication. b. The reviewer is right. We performed flow cytometry against CD80, CD86, HLA-DP, and -DQ. The new results has been added to the main manuscript. Activated WJ-MSCs are not characterized by expression of co-stiumulatory molecules, only by the elevated expression of HLA-class II. We have included this data in the supplementary file (Table S2) and also in the main manuscript in Results section 2.2. Impact of IFN-γ Priming On WJ-MSCs Characteristics “Immunophenotypic analysis indicated the upregulation of HLA-DR, -DQ, -DP and CD10 expression (63.9 ± 9.8 %, 62.1 ± 4.1 %, 61.4 ± 9.8 %,and 91.4 ± 4.2 %, respectively) in IFN-γ primed WJ-MSCs, whereas no altered expression levels regarding theCD340 and CD49a were observed between IFN-γ primed and non-primed WJ-MSCs (Figure 2 and Table S2).” However, we didn’t perform any immunophenotypic analysis regarding the ICAM-1 (CD54) because we didn’t have in the laboratory this mAb. It would take to us a great time to order and to perform the flow cytometry assay, that will exheed enormously the time that is required for the experiment performance and results analysis, beyond the given time by the journal. In a future setup also the ICAM-1 will be included. c. The reviewer is right. In the introduction section we have stated that IL-1b is anti-inflammatory cytokine. We intended to mention that IL-1Ra has anti-inflammatory properties and we have performed this correction that has been indicated with track changes. Regarding the IL-6, recent evidence support the pleiotropic action of this cytokine rather than only its inflammatory. Also, from our experimental results we noticed that IL-6, TGF-β and IL-13 are the key proteins that play crucial role for the induction of the other anti-inflammatory molecules. The was clearly stated in the discussion section. “. Indeed, it has been shown that IL-6 exert a pleotropic action, regulating possibly the balance between pro and anti-inflammatory cytokine production [54]. Moreover, supporting evidence is provided by the fact that IL-6-deficient mice failed to produce proper anti-inflammatory responses, an event which contributes to autoimmune disorders occurrence [54–56]. IL-6, produced by IFN-γ primed WJ-MSCs, can act also in autocrine manner, influencing the production of other anti-inflammatory cytokines and favoring the Th2 shifting.” We have performed revisions in the introduction and discussion sections to show the pleiotropic action of IL-6 rather than its inflammatory,to avoid any misunderstandings to the readers of this work. We have included also to the discussion section the following sentence. “Beyond the above cytokines, also elevated levels of IL-6 was evidenced by the IFN-γ primed WJ-MSCs” |
|
Specific concerns 1- The dot plot of Figure 1 panel C should contain the same number of events. This is not the case, evidently. |
|
Author’s Response |
|
All dot plots involve approximately 100.000 events analysed with the flow cytometer BD Canto II. However, we performed again the evaluation of MSCs markers with the flow cytometry and the new dot plots have been added to the figures. |
|
Reviewer’s Comment |
|
2- The finding/observation that IFN-γ primed-WJ-MSCs show some increase in intracellular vesicles should be demonstrated by conventional and/or advanced intracellular microscopy examination. The data shown does not allow to state what the authors stated. |
|
Author’s Response |
|
The reviewer is right. To state the presence of intracellular vesicles advanced intracellular microscopy should be performed. However, by using light microscopy we were able to observe these vesicles and we do think that is an important observation regarding the morphological changes which depicts the impact of IFN-γ on MSCs. Further images indicating the presence of vesicles in activated MSCs compared to non activated has been imported to supplementary file. |
|
Reviewer’s Comment |
|
3- The upregulation of HLA-II should be shown in all the IFN-γ primed-WJ-MSCs and corresponding controls. The example presented in panel C is informative, but the strength of this paper is in the number of samples analyzed, and this should be stressed. |
|
Author’s Response |
|
In the supplementary file, table S2 the percentage of CD markers expression in non-primed and IFN-γ primed WJ-MSCs for 20 samples has been provided alongside with the statistics analysis. Moreover, as supplemtary material we provided the measurements for each sample of 20 samples with flow cytometry, to show the elevated expression of HLA class II and other CD markers altered expression between IFN-γ primed and non primed (control) WJ-MSCs. |
|
Reviewer’s Comment |
|
4- The cell death of IFN-γ primed-WJ-MSCs should be shown with an additional methods that indeed identify dying/apoptotic cells. |
|
Author’s Response |
|
The evaluation of cell death of IFN-γ primed WJ-MSCs has been performed using the Trypan Blue exclusion dye, determination of ATP and ADP/ATP ratio. Based on the ISO 22859:2022 and ISO/AWI 24651 for MSCs biobanking, the viability method which is indicated for the determination of MSCs viability is the trypan blue. However, we further performed flow cytometry using the 7AAD method to confirm further our results. The flow cytometry regarding the 7AAD results has been included in the supplementary file of this manuscript. |
|
Reviewer’s Comment |
|
5- MLR is performed (apparently) by mixing T cells separated from two cord blood donors. The T cells have been separated by FACS as indicated in the M&M section. The method of separation may trigger the CD3/TCR complex. A negative selection system of separation should be used to avoid the CD3-mediated signalling. The absence of monocytes could impair/alter the MLR response. Further, the presence of antigen-presenting cells is essential for an appropriate MLR. Please clarify. The analysis after 72 hours of MLR is quite short. Usually, the MLR can give a good response in 5-7 days of stimulation. Please explain. |
|
Author’s Response |
|
We would like to thank the reviewer for this valuable comment. We performed again the MLR with the following revisions that can be found in the Method section of the submitted manuscript.
“Mixed Lymphocyte Reaction (MLR) was performed to assess the immunosuppressive properties of IFN-γ primed and non-primed WJ-MSCs, through direct and indirect contact. Mononuclear cells were isolated from 5 ml of two different CBUs, using the ficoll-paque method (Cytiva Ficoll-Paque™ PLUS Media, Sigma-Aldrich, Darmstadt, Germany), as has been previously described [74]. An equal number of allogeneic CB-MNCs (obtained from a different donor), referred to as responders, was isolated, followed by magnetic separation of T cells using the Pan T Cell Isolation Kit (Miltenyi, Biotec, Bergisch Gladbach, Germany) to avoid activation of T cells. For the direct contact 5 x 103 CB-MNCs, referred as stimulators, were seeded in a 24-well plate (Costar, Corning Life, Canton, MA, United States) and treated with mitomycin 25 μg/mL (Sigma-Aldrich, Darmstadt, Germany) for 30 min at 37 °C. An equal number of allogeneic CB-T cells (obtained from a different donor), referred to as responders, was added to the culture. Then, 5 x 103 of either IFN-γ primed or non-primed WJ-MSCs were applied to further assess their effectiveness in halting lymphocyte proliferation. In each well, 1 ml of cultured medium was used. For the indirect contact, the same number of stimulator and responder cells were seeded on the insert (with a pore size 0.4 μm). IFN-γ primed WJ-MSCs or non-primed WJ-MSCs, were seeded on the bottom of the 24-well plate. The culture medium used in the above tasks consisted of a-MEM with 15% FBS, 1% P-S and 1% L-glutamine. The total number of CB-T cells was counted with the Countess 3 automated cell counter (Thermo Fischer Scientific, Waltham, United States), as earlier described in this study. After 6 days (144 hrs), the CB-T cells were counted again, and the proliferation index was determined. In the above assays, as a positive control group, 5 x 103 CB-T cells treated with 5 μg/ ml phytohemagglutinin (PHA, Sigma-Aldrich, Darmstadt, Germany), and as a negative control, 5 x 103 unstimulated CB-T cells, were applied.”
Also the image E in figure 2 has been revised accordingly, including the results of the revised method. We think with this revision we have addressed all the concerns of the reviewers. |
|
Reviewer’s Comment |
|
6- It is unusual that the PHA response is almost superimposable to MLR (panel E, figure 2) tested at 72 hours. Usually, PHA can give a 10-20 fold stronger response compared to MLR. Indeed, the cells that can respond to MLR are usually 1 out of 10,0, while the large majority of T cells can respond to PHA (almost 100 out of 100). |
|
Author’s Response |
|
Again we would like to thank the reviewer for this valuable comment. Indeed the MLR was not appropriately performed and that’s why we had apparently similar number of activated responders cells in PHA group (control) and MLR group. For this purpose, we repeated the whole MLR procedure as previously mentioned and the new results have been imported in figure 2. In addition, detailed information regarding the new MLR assay for the different group numbers can be found in Table S3 and S4 in supplementary material. |
|
Reviewer’s Comment |
|
7- It would be of interest to define whether the frequency of the different HLA class I and II alleles is in line with the frequency of these alleles in the whole population (not only in the WJ-MSCs). |
|
Author’s Response |
|
We have clearly indicate in the discussion section, that the determined HLA alleles of the WJ-MSCs of this study corresponded to the most frequent HLA class I and II alleles in the Greek population. These data has been obtained from a previous published work of my group, entitled “Frequency distribution of HLA class I and II alleles in Greek population and their significance in orchestrating the National Donor Registry Program” published in the International Journal of Immunogenetics. In this work the determination of the HLA alleles, frequencies and haplotypes was based on healthy individuals of different regions from Greece. So the HLA alleles frequencies of the WJ-MSCs of the submitted study was similar to the frequencies obtained from my previous work and that’s why it was stated the following in the discussion section. “Trying to answer this question, high-resolution HLA typing showed that the WJ-MSCs used in this study, characterized by the most common HLA alleles in terms of frequency in the Greek population [61].” We do think that no additional revision it is required here, because there is clear statement regarding the determined frequencies of the HLA alleles of WJ-MSCs of this study. |
|
Reviewer’s Comment |
|
8- The functional relevance of the upregulation of HLA-II as well as HLA-I on IFN-γ primed-WJ-MSCs should be tested. |
|
Author’s Response |
|
To evaluate the functional relevance of the upregulation of HLA class I and II alleles is not any easy task. In the beginning resting non activated MSCs do express HLA class I to avoid the allorecognition by the NK cells. In addition, MLR or T cell cytotoxicity assay cannot be used to show the functional relevance of HLA class II. IFN-γ primed MSCs do express elevated levels of HLA class II, but also secrete high levels of immunomodulatory molecules to tolerate the overactivated immune responses. In this way, we think to acquire clear evidence regarding the function relevance of the upregulated HLA class I and II, this should be tested in animal models. Animal models, will be designed as future step of this study. Tp use animal models will take a great time to produce them and to analyse them using assays like flow cytometry, mass cytometry and histology, which eventually will take over of 1 year of experiments. Also, we have the evidence from the study of Bezstarosti et . al. Front Genet. 2024 Sep 27;15:1436194, that indicated the presence of DSAs and eventually the sensitization of 3 kidney transplanted patients. More work is required in this field to have safe conclusions regarding the relevance of HLA class II mostly. To use animal models to test the functional relevance of HLA class I and II, is a task that we do think to perform it in the near future. However, to test the functional relevance of the HLA class I and II, is beyond the limits of this study and will eventually take more than 1 year to have the initial results. |
Beyond the above revisions, the whole manuscript has been checked for any grammar and phrase errors by a native English speaker and revised accordingly as has been suggested by the reviewers.
We think that with the performed revisions, the quality of the manuscript has been increased and can be further processed to the next step of the publication process.
We remain at your disposal if anything else is required by our side.
Yours sincerely,
Panagiotis Mallis, MSc, PhD
GMP Lab Technical Supervisor
Hellenic Cord Blood Bank,
Biomedical Research Foundation Academy of Athens,
4 Soranou Efessiou Street,
11527 Athens, Greece
tel:+302106597340, mob:+306971616467
e-mail: pmallis@bioacademy.gr

Reviewer 4 Report
Comments and Suggestions for Authors
The manuscript is of high quality and the topic will draw attention of readers withouth any doubt. The ability of MSC to express HLA (especially HLA-DR) in response to proinflammation stimuli and as a consequence to provoke immune responce to donor cells is underestimated in many studies and clinical research protocols.
In general, I conseder the MS as a one of high quality that requires some minor improvements before publication
- The ability of fibroblasts to express human leukocyte antigen (HLA) molecules, particularly when stimulated by interferons, with IFN-γ inducing HLA-DR (a Class II molecule) and increasing already present HLA-A and B (Class I) expression, is a well-known fact. Could you please discuss why the cells used in the study are MSC but not fibroblasts. In fact, fibroblasts' immunophenotype and morphology are very similar to MSC
- The main text should be double-checked for typos. Some sentences are in italic whithout any specific meaning (Discussion section, page 10). IFN-gamma is typed as Ifn-Γ in Materials and Methods. WJ-MSC are sometimes referred as Wj-MSC, MSCs as Mscs etc.
Author Response
Dear Reviewer 4,
We would like to thank the reviewer for the comments performed to our manuscript. We have revised the manuscript based on the reviewer’s comments and below you can find our step by step responses. All changes performed in the revised manuscript have been indicated with the track changes tool of the Microsoft word tool.
|
General Comments |
|
The manuscript is of high quality and the topic will draw attention of readers withouth any doubt. The ability of MSC to express HLA (especially HLA-DR) in response to proinflammation stimuli and as a consequence to provoke immune responce to donor cells is underestimated in many studies and clinical research protocols. In general, I conseder the MS as a one of high quality that requires some minor improvements before publication |
|
Author’s Response |
|
We gratefully thank the reviewer for the comprehensive peer-review performed in our submitted manuscript. This work is very important for our team to be published in the IJMS, and reflects our efforts over the last year to demonstrate that primed MSCs from an inflammatory signal can upregulate the HLA class II expression, which is should be taken into account by the physicians when these stem cells are used as third party adoptive immunotherapy. The results presented herein provide further evidence for the greater understanding of the immunobiology of MSCs and how can we manipulate them, in order to provide a safe alternative therapeutic strategy for patients suffering from immune related disorders. |
|
Reviewer’s Comments |
|
1. The ability of fibroblasts to express human leukocyte antigen (HLA) molecules, particularly when stimulated by interferons, with IFN-γ inducing HLA-DR (a Class II molecule) and increasing already present HLA-A and B (Class I) expression, is a well-known fact. Could you please discuss why the cells used in the study are MSC but not fibroblasts. In fact, fibroblasts' immunophenotype and morphology are very similar to MSC. |
|
Author’s Response |
|
In our study, we examined the fact that primed MSCs by an inflammatory stimulus such as IFN-γ can lead to the upregulation of HLA class II. MSCs are stem cells which have attracted the interest of the scientific community regarding the ability to differentiate to other cell types, hence they are globally used in tissue engineering and regenerative medicine approaches. Beyond the regenerative properties, MSCs may represent the only stem cell population in the human body which has a great association with the immune system. MSCs can be stimulated by inflammatory cues exerted by the macrophages, communicate with them and uptake processed antigenic epitopes to be exposed either with the HLA class I or II molecules. In addition, primed MSCs when located to a highly inflammatory microenvironment can tolerate the overactivated immune responses through specific cellular communication, and specifically either through direct or indirect contact (secretion of anti-inflammatory biomolecules.) This ability of MSCs is of great importance and for this purpose nowdays they are utilized to a great number of clinical trials focused on immune-related disorders such as autoimmune diseases, GvHD, SARS-CoV-2. MSCs with the latest EU Regulation (Regulation (EC) No 1394/2007) are considered ATMPs, which can represent an alternative therapeutic solution for severely conditioned patients suffering from the aforementioned disorders. The MSCs for a long time considered immunoprevileged cells, which cannot induce host immune responses against them, therefore there are registered clinical trials indicating the beneficial use of third party pooled MSCs. MSCs can be derived from different sources of the human body including the bone marrow, adipose tissue and umbilical cord. Hoever, adult derived MSCs and dependent to the patients condition, may be characterized by altered immunomodulatory properties, proliferation and differentiation potential compared to the fetal derived or even those derived from healthy individuals. Given that MSCs considered as immune privileged cells, pooled strategies of those stem cells were developed. Katerina Le Blanc was the first who noticed the intracellular expression of HLA class II in MSCs, however since then limited evidence in the literature explored this fact. In our study we demonstrated the upregulation of HLA class II upon IFN-γ licensing, which may be associated with the immunomodulatory action. Indeed resting MSCs do not express HLA class II, however when MSCs activated by inflammatory cues, HLA class II are upregulated and expressed. In this way, when MSCs are administrated in patients suffering from immune-related disorders, this inflammatory stimulus will cause the priming of these cells and the expression of the HLA class II. In a situation like this, the MSCs may elicit rapid host immune response against them (if characterized by histocompatibility mismatches) which result to rapid clearance of the MSCs before exerting their immunoregulatory properties. In this study we explore this possibility. Specifically, the HLA histocompatibility of MSCs between donor and recipient should importantly considered in the selection of the cells, in order to avoid any sensitization adverse events by the host. All these details are clearly presented in the submitted manuscript. On the other hand, fibroblasts are not stem cells, constitutively express HLA class I and II molecules. The only common between MSCs and fibroblasts are actually their spindle shape morphology. Fibroblasts are not characterized by immunoregulatory properties and differentiation to other cellular types. Therefore fibroblasts cannot be used as an alternative ATMP therapy in immune related disorders. In our study we used well defined MSCs according to the criteria of ISCT to perform all experimental series. Based on these MSCs, are characterized by totally different immunophenotype, proliferation, differentiation and immunomodulatory properties compared to fibroblasts. Therefore MSCs can be used as potential therapeutic strategy in severely conditioned patients, whereas fibroblasts no. |
|
Reviewer’s Comments |
|
2. The main text should be double-checked for typos. Some sentences are in italic whithout any specific meaning (Discussion section, page 10). IFN-gamma is typed as Ifn-Γ in Materials and Methods. WJ-MSC are sometimes referred as Wj-MSC, MSCs as Mscs etc. |
|
Author’s Response |
|
The reviewer is right. The whole manuscript has been double checked for any typos. It seemed some typos were introduced during the conversion of the manuscript to the IJMS template during submission. All these typos were corrected. Also any grammar or phrase errors were revised in the whole manuscript by a native English speaker of our team. |
We think that with the performed revisions, the quality of the manuscript has been increased and can be further processed to the next step of the publication process.
We remain at your disposal if anything else is required by our side.
Yours sincerely,
Panagiotis Mallis, MSc, PhD
GMP Lab Technical Supervisor
Hellenic Cord Blood Bank,
Biomedical Research Foundation Academy of Athens,
4 Soranou Efessiou Street,
11527 Athens, Greece
tel:+302106597340, mob:+306971616467
e-mail: pmallis@bioacademy.gr

Round 2
Reviewer 2 Report
Comments and Suggestions for Authors
The authors have adequately addressed all concerns. Therefore, I recommend this paper for publication in IJMS.
Reviewer 3 Report
Comments and Suggestions for Authors
The authors replied to the reviewer's questions. The large majority of these responses are presented in the supplementary materials. Unfortunately, it was not possible for this reviewer to view the content of these supplementary data because the PDF was illegible.
I leave the Editor to check the consistency between the replies and the supplementary material section to accept this manuscript for publication,